# CloudLSTM: A Recurrent Neural Model for Spatiotemporal Point-cloud Stream Forecasting

## Abstract

This paper introduces CloudLSTM, a new branch of recurrent neural models tailored to forecasting over data streams generated by geospatial point-cloud sources. We design a Dynamic Point-cloud Convolution (DConv) operator as the core component of CloudLSTMs, which performs convolution directly over point-clouds and extracts local spatial features from sets of neighboring points that surround different elements of the input. This operator maintains the permutation invariance of sequence-to-sequence learning frameworks, while representing neighboring correlations at each time step – an important aspect in spatiotemporal predictive learning. The DConv operator resolves the grid-structural data requirements of existing spatiotemporal forecasting models and can be easily plugged into traditional LSTM architectures with sequence-to-sequence learning and attention mechanisms. We apply our proposed architecture to two representative, practical use cases that involve point-cloud streams, i.e., mobile service traffic forecasting and air quality indicator forecasting. Our results, obtained with real-world datasets collected in diverse scenarios for each use case, show that CloudLSTM delivers accurate long-term predictions, outperforming a variety of neural network models.

## 1 Introduction

Point-cloud stream forecasting aims at predicting the future values and/or locations of data streams generated by a geospatial point-cloud $\mathcal{S}$, given sequences of historical observations (Shi & Yeung, 2018). Example data sources include mobile network antennas that serve the traffic generated by ubiquitous mobile services at city scale (Zhang et al., 2019b), sensors that monitor the air quality of a target region (Cheng et al., 2018), or moving crowds that produce individual trajectories. Unlike traditional spatiotemporal forecasting on grid-structural data, like precipitation nowcasting (Shi et al., 2015) or video frame prediction (Wang et al., 2018), point-cloud stream forecasting needs to operate on geometrically scattered sets of points, which are irregular and unordered, and encapsulate complex spatial correlations. While vanilla Long Short-term Memories (LSTMs) have modest abilities to exploit spatial features (Shi et al., 2015), convolution-based recurrent neural network (RNN) models, such as ConvLSTM (Shi et al., 2015) and PredRNN++ (Wang et al., 2018), are limited to modeling grid-structural data, and are therefore inappropriate for handling scattered point-clouds.

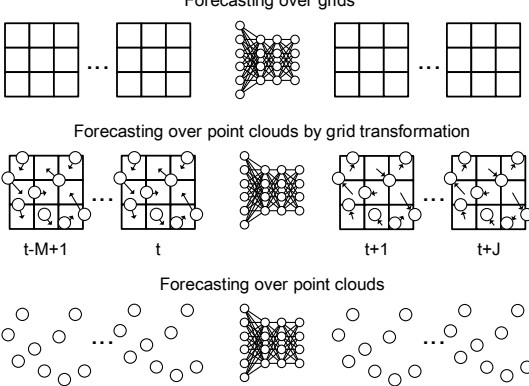

Figure 1: Different approaches to geospatial data stream forecasting: predicting over input data streams that are inherently grid-structured, e.g., video frames using ConvLSTMs (top); mapping of point-cloud input to a grid, e.g., mobile network traffic collected at different antennas in a city, to enable forecasting using existing neural network structures (middle); forecasting directly over point-cloud data streams using historical information (as above, but without pre-processing), as proposed in this paper (bottom).

Leveraging the location information embedded in such irregular data sources, so as to learn important spatiotemporal features, is in fact challenging. Existing approaches that tackle the point-cloud stream forecasting problem can be categorized into two classes, both of which bear significant shortcomings: *(i)* methods that transform point-clouds into data structures amenable to processing with mature solutions, e.g., grids as exemplified in Fig.1 (Zhang et al., 2019a); and *(ii)* models that ignore the exact locations of each data source and inherent spatial correlations (Liang et al., 2018). The transformations required by the former not only add data preprocessing overhead, but also introduce spatial displacements that distort relevant correlations among points (Zhang et al., 2019a). On the other hand, the latter are largely location-invariant, while recent literature suggests spatial correlations should be revisited over time, to suit series prediction tasks (Shi et al., 2017). In essence, overlooking dynamic spatial correlations will lead to modest forecasting performance.

**Contributions.** In this paper, we introduce Convolutional Point-cloud LSTMs (CloudLSTMs), a new branch of recurrent neural network models tailored to geospatial point-cloud stream forecasting. The CloudLSTM builds upon a Dynamic Point-cloud Convolution (DConv) operator, which takes raw point-cloud streams (both data time series and spatial coordinates) as input, and performs dynamic convolution over these, to learn spatiotemporal features over time, irrespective of the topology and permutations of the point-cloud. This eliminates the need for the data preprocessing described above, and so avoids the distortions thereby introduced. The proposed CloudLSTM takes into account the locations of each data source and performs *dynamic positioning* at each time step, to conduct a deformable convolution operation over point-clouds (Dai et al., 2017). This allows revising the spatiotemporal correlations and the configuration of the data points over time, and guarantees the location-invariant property is met at different steps. Importantly, the DConv operator is flexible, as it can be easily plugged into existing neural network models for different purposes, such as RNNs, LSTMs, sequence-to-sequence (Seq2seq) learning (Sutskever et al., 2014), and attention mechanisms (Luong et al., 2015).

We evaluate our proposed architectures on four benchmark datasets, covering two spatiotemporal point-cloud stream forecasting applications: *(i)* antenna-level forecasting of data traffic generated by mobile services (Zhang & Patras, 2018; Bega et al., 2019), leveraging metropolitan-scale mobile traffic measurements collected in two European cities for 38 popular mobile apps; and *(ii)* forecasting six air quality indicators on two city clusters in China (Zheng et al., 2015). These tasks represent important use cases of geospatial point-cloud stream forecasting. We combine our CloudLSTM with Seq2seq learning and an attention mechanism, then undertake a comprehensive evaluation on all datasets. The results demonstrate that our architecture can deliver precise long-term point-cloud stream forecasting in different settings, outperforming 12 different baseline neural network models in terms of four performance metrics, without any data preprocessing requirements. To the best of knowledge, the proposed CloudLSTM is the *first dedicated neural architecture* for spatiotemporal forecasting that *operates directly on point-cloud streams*.

## 2 RELATED WORK

**Spatiotemporal Forecasting.** Convolution-based RNN architectures have been widely employed for spatiotemporal forecasting, as they simultaneously capture spatial and temporal dynamics of the input. Shi et al., incorporate convolution into LSTMs, building a ConvLSTM for precipitation nowcasting (Shi et al., 2015). This approach exploits spatial information, which in turn leads to higher prediction accuracy. The ConvLSTM is improved by constructing a subnetwork to predict state-to-state connections, thereby guaranteeing location-variance and flexibility of the model (Shi et al., 2017). PredRNN (Wang et al., 2017) and PredRNN++ (Wang et al., 2018) evolve the ConvLSTM by constructing spatiotemporal cells and adding gradient highway units. These improve long-term forecasting performance and mitigate the gradient vanishing problem in recurrent architectures. Although these solution work well for spatiotemporal forecasting, they can not be applied directly to point-cloud streams, as they require point-cloud-to-grid preprocessing (Zhang et al., 2019a).

**Feature Learning on Point-clouds.** Deep neural networks for feature learning on point-cloud data are advancing rapidly. PointNet performs feature learning and maintains input permutation invariance (Qi et al., 2017a). PointNet++ upgrades this structure by hierarchically partitioning point-clouds and performing feature extraction on local regions (Qi et al., 2017b). VoxelNet employs voxel feature encoding to limit inter-point interactions within a voxel (Zhou & Tuzel, 2018). This effectively projects cloud-points onto sub-grids, which enables feature learning. Li et al., generalize the convolution operation on point-clouds and employ $\mathcal{X}$-transformations to learn the weights and

permutations for the features (Li et al., 2018). Through this, the proposed PointCNN leverages spatial-local correlations of point clouds, irrespective of the order of the input. Notably, although these architectures can learn spatial features of point-clouds, they are designed to work with static data, thus have limited ability to discover temporal dependencies.

## 3    CONVOLUTIONAL POINT-CLOUD LSTM

Next, we describe in detail the concept and properties of forecasting over point cloud-streams. We then introduce the DConv operator, which is at the core of our proposed CloudLSTM architecture. Finally, we present CloudLSTM and its variants, and explain how to combine CloudLSTM with Seq2seq learning and attention mechanisms, to achieve precise forecasting over point-cloud streams.

### 3.1    FORECASTING OVER POINT-CLOUD STREAMS

We formally define a point-cloud containing a set of $N$ points, as $\mathcal{S} = \{p_1, p_2, \cdots, p_N\}$. Each point $p_n \in \mathcal{S}$ contains two sets of features, i.e., $p_n = \{\nu_n, \varsigma_n\}$, where $\nu_n = \{v_n^1, \cdots, v_n^H\}$ are value features (e.g., mobile traffic measurements, air quality indexes, etc.) of $p_n$, and $\varsigma_n = \{c_n^1, \cdots, c_n^L\}$ are its $L$-dimensional coordinates. At each time step $t$, we may obtain $U$ different channels of $\mathcal{S}$ by conducting different measurements denoted by $\mathcal{S}_t^v = \{\mathcal{S}_t^1, \cdots, \mathcal{S}_t^U\}$, $\mathcal{S}_t^v \in \mathbb{R}^{U \times N \times (H+L)}$. Here, different $U$ resemble the RGB channels in images. We can then formulate the $J$-step point-cloud stream forecasting problem, given $M$ observations, as:

$$\widehat{\mathcal{S}}_{t+1}^v, \cdots, \widehat{\mathcal{S}}_{t+J}^v = \underset{\mathcal{S}_{t+1}^v, \cdots, \mathcal{S}_{t+J}^v}{\mathrm{argmax}} \; p(\mathcal{S}_{t+1}^v, \cdots, \mathcal{S}_{t+J}^v | \mathcal{S}_t^v, \cdots, \mathcal{S}_{t-M+1}^v). \tag{1}$$

Note that, in some cases, each point's coordinates may be unchanged, since the data sources are deployed at fixed locations. An ideal point-cloud stream forecasting model should embrace five key properties, similar to other point-cloud applications and spatiotemporal forecasting problems (Qi et al., 2017a; Shi et al., 2017):

*(i)* **Order invariance:** A point cloud is usually arranged without a specific order. Permutations of the input points should not affect the output of the forecasting (Qi et al., 2017a).

*(ii)* **Information intactness:** The output of the model should have exactly the same number of points as the input, without losing any information, *i.e.*, $N_{\mathrm{out}} = N_{\mathrm{in}}$.

*(iii)* **Interaction among points:** Points in $\mathcal{S}$ are not isolated, thus the model should be able to capture local dependencies among neighboring points and allow interactions (Qi et al., 2017a).

*(iv)* **Robustness to transformations:** The model should be robust to correlation-preserving transformation operations on point-clouds, e.g., scaling and shifting (Qi et al., 2017a).

*(v)* **Location variance:** The spatial correlations among points may change over time. Such dynamic correlations should be revised and learnable during training (Shi et al., 2017).

In what follows, we introduce the Dynamic Point Cloud Convolution (DConv) operator as the core module of the CloudLSTM, and explain how DConv satisfies the aforementioned properties.

### 3.2    DYNAMIC CONVOLUTION OVER POINT CLOUD

The Dynamic Point Cloud Convolution operator (DConv) generalizes the ordinary convolution on grids. Instead of computing the weighted summation over a small receptive field for each anchor point, DConv does so on point-clouds, while inheriting desirable properties of the ordinary convolution operation. The vanilla convolution takes $U_{\mathrm{in}}$ channels of 2D tensors as input, and outputs $U_{\mathrm{out}}$ channels of 2D tensors of smaller size (if without padding). Similarly, the DConv takes $U_{\mathrm{in}}$ channels of a point-cloud $\mathcal{S}$, and outputs $U_{\mathrm{out}}$ channels of a point-cloud, but with the same number of elements as the input, to ensure the *information intactness* property *(ii)* discussed previously. For simplicity, we denote the $i^{th}$ channel of the input set as $\mathcal{S}_{in}^i$ and the $j^{th}$ channel of the output as $\mathcal{S}_{\mathrm{out}}^j$. Both $\mathcal{S}_{in}^i$ and $\mathcal{S}_{\mathrm{out}}^j$ are 3D tensors, of shape $(N, (H+L), U_{\mathrm{in}})$ and $(N, (H+L), U_{\mathrm{out}})$ respectively.

We also define $\mathcal{Q}_n^{\mathcal{K}}$ as a subset of points in $\mathcal{S}_{in}^i$, which includes the $\mathcal{K}$ nearest points with respect to $p_n$ in the Euclidean space, i.e., $\mathcal{Q}_n^{\mathcal{K}} = \{p_n^1, \cdots, p_n^k, \cdots, p_n^{\mathcal{K}}\}$, where $p_n^k$ is the $k$-th nearest point to $p_n$ in the set $\mathcal{S}_{in}^i$. Note that $p_n$ itself is included in $\mathcal{Q}_n^{\mathcal{K}}$ as an anchor point, i.e., $p_n \equiv p_n^1$. Recall that each $p_n \in \mathcal{S}$ contains $H$ value features and $L$ coordinate features, i.e., $p_n = \{\nu_n, \varsigma_n\}$, where $\nu_n = \{v_n^1, \cdots, v_n^H\}$ and $\varsigma_n = \{c_n^1, \cdots, c_n^L\}$. Similar to the vanilla convolution operator, for each $p_n$

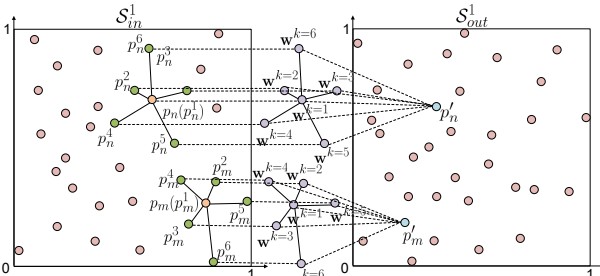

Figure 2: Illustration of the DConv operator, with a single input channel and $\mathcal{K} = 6$ neighbors. For every $p \in \mathcal{S}_{in}^1$, DConv weights its $\mathcal{K}$ neighboring set $\mathcal{Q}_n^{\mathcal{K}} = \{p_n^1, \cdots, p_n^6\}$ to produce values and coordinate features for $p_n' \in \mathcal{S}_{out}^1$. Here, each $\mathbf{w}^k$ is a set of weights $w$ with index $k$ (i.e., $k$-th nearest neighbor) in Eq. 2, shared across different $p$.

in $\mathcal{S}_{in}^i$, the DConv sums the element-wise product over all features and points in $\mathcal{Q}_n^{\mathcal{K}}$, to obtain the values and coordinates of a point $p_n'$ in $\mathcal{S}_{out}^j$. Note that we assume the value features are related to their positions at the previous layer/state, to better exploit the dynamic spatial correlations. Therefore, we aggregate coordinate features $c(p_n^k)_i^l$ when computing the value features $v_{n,j}^{h'}$. The mathematical expression of the DConv is thus:

$$v_{n,j}^{h'} = \sum_{i \in U_{in}} \sum_{p_n^k \in \mathcal{Q}_n^{\mathcal{K}}} \left( \sum_{h \in H} w_{i,j}^{h,h',k} v(p_n^k)_i^h + \sum_{l \in L} w_{i,j}^{(H+l),h',k} c(p_n^k)_i^l \right) + b_j,$$

$$c_{n,j}^{l'} = \sigma \left( \sum_{i \in U_{in}} \sum_{p_n^k \in \mathcal{Q}_n^{\mathcal{K}}} \left( \sum_{h \in H} w_{i,j}^{h,l',k} v(p_n^k)_i^h + \sum_{l \in L} w_{i,j}^{(H+l),l',k} c(p_n^k)_i^l \right) + b_j \right),$$

$$\mathcal{S}_{out}^j = (p_1', \cdots, p_N') \tag{2}$$

$$= \left( \left( (v_1^{1'}, \cdots, v_1^{H'}), (c_1^{1'}, \cdots, c_1^{L'}) \right), \cdots, \left( (v_N^{1'}, \cdots, v_N^{H'}), (c_N^{1'}, \cdots, c_N^{L'}) \right) \right).$$

In the above, we define learnable weights $\mathcal{W}$ as 5D tensors with shape $(U_{in}, \mathcal{K}, (H+L), (H+L), U_{out})$. The weights are shared across different anchor points in the input map. Each element $w_{i,j}^{m,m',k} \in \mathcal{W}$ is a scalar weight for the $i$-th input channel, $j$-th output channel, $k$-th nearest neighbor of each point corresponding to the $m$-th value and coordinate features for each input point, and $m'$-th value and coordinate features for output points. Similar to the convolution operator, we define $b_j$ as a bias for the $j$-th output map. In the above, $h$ and $h'$ are the $h^{(\prime)}$-th value features of the input/output point set. Likewise, $l$ and $l'$ are the $l^{(\prime)}$-th coordinate features of the input/output. $\sigma(\cdot)$ is the sigmoid function, which limits the range of predicted coordinates to $(0, 1)$, to avoid outliers. Before feeding them to the model, the coordinates of raw point-clouds are normalized to $(0, 1)$ by $\varsigma = (\varsigma - \varsigma_{min})/(\varsigma_{max} - \varsigma_{min})$, on each dimension. This improves the transformation robustness of the operator.

The $\mathcal{K}$ nearest points can vary for each channel at each location, because the channels in the point-cloud dataset may represent different types of measurements. For example, channels in the mobile traffic dataset are related to the traffic consumption of different mobile apps, while those in the air quality dataset are different air quality indicators ($SO_2$, CO, etc.). The spatial correlations will vary between different measurements (channels), due to human mobility. For instance, more people may use Facebook at a social event, but YouTube traffic may be less significant in this case. This will be reflected by the data consumption of each app. The same applies to air quality indicators affected by vehicle movement and factory working times. We want these spatial correlations to be learnable, so we do not fix the $\mathcal{K}$ nearest neighbors across channels, but encourage each channel to find the best neighbor set. This is also a contribution of the CloudLSTM, which helps improve the forecasting performance.

We provide a graphical illustration of DConv in Fig. 2. For each point $p_n$, the DConv operator weights its $\mathcal{K}$ nearest neighbors across all features, to produce the values and coordinates in the next layer. Since the permutation of the input neither affects the neighboring information nor the ranking of their distances for any $\mathcal{Q}_n^{\mathcal{K}}$, DConv is a symmetric function whose output does not depend on the input order. This means that the property *(i)* discussed in Sec. 3.1 is satisfied. Further, DConv is performed on every point in set $\mathcal{S}_{in}^i$ and produces exactly the same number of features and points for its output; property *(ii)* is therefore naturally fulfilled. In addition, operating over a neighboring point set, irrespective of its layout, allows to capture local dependencies and improve the robustness

to global transformations (e.g., shifting and scaling). The normalization over the coordinate features further improves the robustness to those transformations, as shown by the proof in Appendix C. This enables to meet the desired properties *(iii)* and *(iv)*. More importantly, DConv learns the layout and topology of the cloud-point for the next layer, which changes the neighboring set $\mathcal{Q}_n^{\mathcal{K}}$ for each point at output $\mathcal{S}_{\text{out}}^j$. This enables the "location-variance" (property *(v)*), allowing the model to perform dynamic positioning tailored to each channel and time step. This is essential in spatiotemporal forecasting neural models, as spatial correlations change over time (Shi et al., 2017). DConv can be efficiently implemented using simple 2D convolution, by reshaping the input map and weight tensor, which can be parallelized easily in existing deep learning frameworks. We detail this in Appendix A and provide a complexity analysis of the DConv operator in Appendix B.

**Relations with PointCNN (Li et al., 2018) and Deformable Convolution (Dai et al., 2017).** The DConv operator builds upon the PointCNN (Li et al., 2018) and deformable convolution neural network (DefCNN) on grids (Dai et al., 2017), but introduces several variations tailored to point-cloud structural data. PointCNN employs the $\mathcal{X}$-transformation over point clouds, to learn the weight and permutation on a local point set using multilayer perceptrons (MLPs), which introduces extra complexity. This operator guarantees the order invariance property, but leads to information loss, since it performs aggregation over points. In our DConv operator, the permutation is maintained by aligning the weight of the ranking of distances between point $p_n$ and $\mathcal{Q}_n^{\mathcal{K}}$. Since the distance ranking is unrelated to the order of the inputs, the order invariance is ensured in a parameter-free manner without extra complexity and loss of information.

Further, the DConv operator can be viewed as the DefCNN (Dai et al., 2017) over point-clouds, with the differences that *(i)* DefCNN deforms weighted filters, while DConv deforms the input maps; and *(ii)* DefCNN employs bilinear interpolation over input maps with a set of continuous offsets, while DConv instead selects $\mathcal{K}$ neighboring points for its operations. Both DefCNN and DConv have transformation modeling flexibility, allowing adaptive receptive fields on convolution.

### 3.3 THE CLOUDLSTM ARCHITECTURE

The DConv operator can be plugged straightforwardly into LSTMs, to learn both spatial and temporal correlations over point-clouds. We formulate the Convolutional Point-cloud LSTM (CloudLSTM) as:

$$
\begin{aligned}
i_t &= \sigma(\mathcal{W}_{si} \circledast \mathcal{S}_t^v + \mathcal{W}_{hi} \circledast H_{t-1} + b_i), \\
f_t &= \sigma(\mathcal{W}_{sf} \circledast \mathcal{S}_t^v + \mathcal{W}_{hf} \circledast H_{t-1} + b_f), \\
C_t &= f_t \odot C_{t-1} + i_t \odot \tanh(\mathcal{W}_{sc} \circledast \mathcal{S}_t^v + \mathcal{W}_{hc} \circledast H_{t-1} + b_c), \\
o_t &= \sigma(\mathcal{W}_{so} \circledast \mathcal{S}_t^v + \mathcal{W}_{ho} \circledast H_{t-1} + b_o), \\
H_t &= o_t \odot \tanh(C_t).
\end{aligned} \tag{3}
$$

Similar to ConvLSTM (Shi et al., 2015), $i_t$, $f_t$, and $o_t$, are input, forget, and output gates respectively. $C_t$ denotes the memory cell and $H_t$ is the hidden states. Note that $i_t$, $f_t$, $o_t$, $C_t$, and $H_t$ are all point cloud representations. $\mathcal{W}$ and $b$ represent learnable weight and bias tensors. In Eq. 3, '$\odot$' denotes the element-wise product, '$\circledast$' is the DConv operator formalized in Eq. 2, and '$\circledast$' a simplified DConv that removes the sigmoid function in Eq. 2. The latter only operates over the gates computation, as the sigmoid functions are already involved in outer calculations (first, second, and fourth expressions in Eq. 3). We show the structure of a basic CloudLSTM cell in the left part of Fig. 3.

We combine our CloudLSTM with Seq2seq learning (Sutskever et al., 2014) and the soft attention mechanism (Luong et al., 2015), to perform forecasting, given that these neural models have been proven to be effective in spatiotemporal modelling on grid-structural data (e.g., (Shi et al., 2015; Zhang et al., 2018)). We show the overall Seq2seq CloudLSTM in the right part of Fig. 3. The architecture incorporates an encoder and a decoder, which are different stacks of CloudLSTMs. The encoder encodes the historical information into a tensor, while the decoder decodes the tensor into predictions. The states of the encoder and decoder are connected using the soft attention mechanism via a context vector (Luong et al., 2015). Before feeding the point-cloud to the model and generating the final forecasting, the data is processed by Point Cloud Convolutional (CloudCNN) layers, which perform the DConv operations. Their function is similar to the word embedding layer in natural language processing tasks (Mikolov et al., 2013), which helps translate the raw point-cloud into tensors and *vice versa*. In this study, we employ a two-stack encoder-decoder architecture, and configure 36 channels for each CloudLSTM cell, as we found that further increasing the number of stacks and channels does not improve the performance significantly.

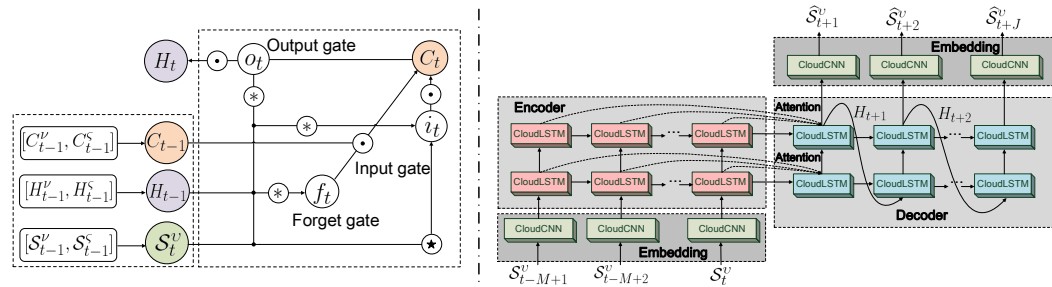

Figure 3: The inner structure of the CloudLSTM cell (left) and the overall Seq2seq CloudLSTM architecture (right). We denote by $(\cdot)^{\nu}$ and $(\cdot)^{\varsigma}$ the value and coordinate features of each input, while these features are unified for gates.

Beyond CloudLSTM, we also explore plugging the DConv into vanilla RNN and Convolutional GRU, which leads to a new Convolutional Point-cloud RNN (CloudRNN) and Convolutional Point-cloud GRU (CloudGRU), as formulated in the Appendix E. The CloudRNN and CloudGRU share a similar Seq2seq architecture with CloudLSTM, except that they do not employ the attention mechanism. We compare their performance in the following section.

## 4    EXPERIMENTS

To evaluate the performance of our architectures, we employ measurement datasets of traffic generated by 38 mobile services and recorded at individual network antennas, and of 6 air quality indicators collected at monitoring stations. We use the proposed CloudLSTM to forecast future mobile service demands and air quality indicators in the target regions. We provide a comprehensive comparison with 12 baseline deep learning models, over four performance metrics. All models considered in this study are implemented using the open-source Python libraries TensorFlow (Abadi et al., 2016) and TensorLayer (Dong et al., 2017). We train all architectures with a computing cluster with two NVIDIA Tesla K40M GPUs. We optimize all models by minimizing the mean square error (MSE) between predictions and ground truth, using the Adam optimizer (Kingma & Ba, 2015).

Next, we first introduce the datasets employed in this study, then discuss the baseline models used for comparison. Finally, we report on the experimental results obtained.

### 4.1    DATASETS AND PREPROCESSING

We conduct experiments on two typical spatiotemporal point-cloud stream forecasting tasks over 2D geospatial environments, with measurements collected in two different scenarios for each use case. Note that, as the data sources have fixed locations in these applications, the coordinate features will be omitted in the final output. However, in different use cases, such as crowd mobility forecasting, the coordinate features would be necessarily included.

**Mobile Traffic Forecasting.** We experiment with large-scale multi-service datasets collected by a major operator in two large European metropolitan areas with diverse topology and size during 85 consecutive days. The data consists of the volume of traffic generated by devices associated to each of the 792 and 260 antennas in the two target cities, respectively. The antennas are non-uniformly distributed over the urban regions, thus they can be viewed as 2D point clouds over space. At each antenna, the traffic volume is expressed in Megabytes and aggregated over 5-minute intervals, which leads to 24,482 traffic snapshots. These snapshots are gathered independently for each of 38 different mobile services, selected among the most popular apps for video streaming, gaming, messaging, cloud services, and social networking. Further details about the dataset can be found in Appendix G.

**Air Quality Forecasting.** Air quality forecasting performance is investigated using a public dataset (Zheng et al., 2015), which comprises six air quality indicators (i.e., PM2.5, PM10, $NO_2$, CO, $O_3$ and $SO_2$) collected by 437 air quality monitoring stations in China, over a span of one year. The monitoring stations are partitioned into two city clusters, based on their geographic locations, and measure data on an hourly basis. The dataset includes 8,760 snapshots in total for each cluster. We conduct experiments on both clusters individually and fill missing data using linear interpolation. The reader is referred to Appendix G for details.

Before feeding to the models, the measurements associated to each mobile service and air quality indicator are transformed into different input channels of the point-cloud $\mathcal{S}$. All coordinate features $\varsigma$

Table 1: The mean±std of MAE, RMSE, PSNR, and SSIM across all models considered, evaluated on two datasets collected in different cities for mobile traffic forecasting.

| Model | City 1 | | | | City 2 | | | |
|---|---|---|---|---|---|---|---|---|
| | MAE | RMSE | PSNR | SSIM | MAE | RMSE | PSNR | SSIM |
| MLP | 4.79±0.54 | 9.94±2.56 | 49.56±2.13 | 0.27±0.12 | 4.59±0.59 | 9.44±2.45 | 50.30±2.28 | 0.33±0.14 |
| CNN | 6.00±0.62 | 11.02±2.09 | 48.93±1.60 | 0.25±0.12 | 5.30±0.51 | 10.05±2.06 | 49.97±1.87 | 0.32±0.14 |
| 3D-CNN | 4.99±0.57 | 9.94±2.44 | 49.74±2.13 | 0.33±0.14 | 5.21±0.48 | 9.97±2.03 | 50.13±1.85 | 0.37±0.16 |
| DefCNN | 6.76±0.81 | 11.72±2.57 | 48.43±1.82 | 0.16±0.08 | 5.31±0.51 | 9.99±2.13 | 49.84±1.87 | 0.32±0.14 |
| PointCNN | 4.95±0.53 | 10.10±2.46 | 49.43±2.06 | 0.27±0.12 | 4.75±0.56 | 9.55±2.32 | 50.17±2.16 | 0.35±0.15 |
| CloudCNN | 4.81±0.58 | 9.91±2.81 | 49.93±2.21 | 0.29±0.11 | 4.68±0.52 | 9.39±2.22 | 50.31±2.03 | 0.36±0.14 |
| LSTM | 4.20±0.66 | 9.58±3.17 | 50.47±3.29 | 0.36±0.10 | 4.32±1.64 | 9.17±3.03 | 50.79±3.26 | 0.42±0.12 |
| ConvLSTM | 3.98±1.60 | 9.25±3.10 | 50.47±3.29 | 0.36±0.10 | 4.09±1.59 | 8.87±2.97 | 51.10±3.33 | 0.42±0.12 |
| PredRNN++ | 3.97±1.60 | 9.29±3.12 | 50.43±3.30 | 0.36±0.10 | 4.07±1.56 | 8.87±2.97 | 51.09±3.34 | 0.42±0.12 |
| PointLSTM | 4.63±0.45 | 9.47±2.55 | 50.02±2.26 | 0.34±0.14 | 4.56±0.54 | 9.26±2.43 | 50.52±2.35 | 0.37±0.15 |
| CloudRNN ($\mathcal{K} = 9$) | 4.08±1.66 | 9.19±3.17 | 50.45±3.23 | 0.32±0.12 | 4.08±1.65 | 8.74±3.03 | 51.10±3.26 | 0.39±0.14 |
| CloudGRU ($\mathcal{K} = 9$) | 3.79±1.59 | 8.90±3.11 | 50.73±3.29 | 0.39±0.10 | 3.90±1.57 | 8.47±2.96 | 51.40±3.33 | 0.45±0.12 |
| CloudLSTM ($\mathcal{K} = 3$) | 3.71±1.63 | 8.87±3.11 | 50.76±3.30 | 0.39±0.10 | 3.86±1.51 | 8.42±2.94 | 51.45±3.32 | 0.46±0.11 |
| CloudLSTM ($\mathcal{K} = 6$) | 3.72±1.63 | 8.91±3.13 | 50.72±3.29 | 0.38±0.10 | 3.84±1.59 | 8.46±2.96 | 51.43±3.33 | 0.45±0.12 |
| CloudLSTM ($\mathcal{K} = 9$) | 3.72±1.62 | 8.88±3.11 | 50.75±3.29 | 0.39±0.10 | 3.89±1.55 | 8.46±2.96 | 51.41±3.32 | 0.46±0.11 |
| Attention CloudLSTM ($\mathcal{K} = 9$) | **3.66±1.64** | **8.82±3.10** | **50.78±3.21** | **0.40±0.11** | **3.79±1.57** | **8.43±2.96** | **51.46±3.33** | **0.47±0.11** |

are normalized to the $(0, 1)$ range. In addition, for the baseline models that require grid-structural input (i.e., CNN, 3D-CNN, ConvLSTM and PredRNN++), the point-clouds are transformed into grids (Zhang et al., 2019a) using the Hungarian algorithm (Kuhn, 1955). The ratio of training plus validation, and test sets is 8:2.

## 4.2 Benchmarks and Performance Metrics

We compare the performance of our proposed CloudLSTM with a set of baseline models, as follows. PointCNN (Li et al., 2018) performs convolution over point-clouds and has been employed for point-cloud classification and segmentation. CloudCNN is an original benchmark we introduce, which stacks the proposed DConv operator over multiple layers for feature extraction from point-clouds. PointLSTM is another original benchmark, obtained by replacing the cells in ConvLSTM with the $\mathcal{X}$-Conv operator employed by PointCNN, which provides a fair term of comparison for other Seq2seq architectures. Beyond these models, we also compare the CloudLSTM with two of its variations, i.e., CloudRNN and CloudGRU, which were introduced in Sec. 3.3. Other baseline models, including MLP (Goodfellow et al., 2016), CNN (Krizhevsky et al., 2012), 3D-CNN (Ji et al., 2013), LSTM (Hochreiter & Schmidhuber, 1997), ConvLSTM (Shi et al., 2015) PredRNN++ (Wang et al., 2018), along with the detailed configuration of all models are discussed in Appendix E.

We quantify the accuracy of the proposed CloudLSTM in terms of Mean Absolute Error (MAE) and Root Mean Square Error (RMSE). Since the mobile traffic snapshots can be viewed as "urban images" (Liu et al., 2015), we also select Peak Signal-to-Noise Ratio (PSNR) and Structural Similarity Index (SSIM) (Hore & Ziou, 2010) to quantify the fidelity of the forecasts and their similarity with the ground truth, as suggested by relevant recent work (Zhang et al., 2017). Details about the metrics are discussed in Appendix F.

For the mobile traffic prediction task, we employ all neural networks to forecast city-scale mobile traffic consumption over a time horizon of $J = 6$ sampling steps, i.e., 30 minutes, given $M = 6$ consecutive past measurements. For RNN-based models, i.e., LSTM, ConvLSTM, PredRNN++, CloudLSTM, CloudRNN, and CloudGRU, we then extend the number of prediction steps to $J = 36$, i.e., 3 hours, to evaluate their long-term performance. In the air quality forecasting use case, all models receive a half day of measurements, i.e., $M = 12$, as input, and forecast indicators in the following 12 hours, i.e., $J = 12$. As for the previous use case, the number of prediction steps are then extended to $J = 72$, or 3 days, for all RNN-based models.

## 4.3 Result on Mobile Traffic Forecasting

We perform 6-step forecasting for 4,888 instances across the test set, and report in Table 1 the mean and standard deviation (std) of each metric. We also investigate the effect of a different number of neighboring points (i.e., $\mathcal{K} = 3, 6, 9$), as well as the influence of the attention mechanism. Observe that RNN-based architectures in general obtain superior performance, compared to CNN-based models and the MLP. In particular, our proposed CloudLSTM, and its CloudRNN, and

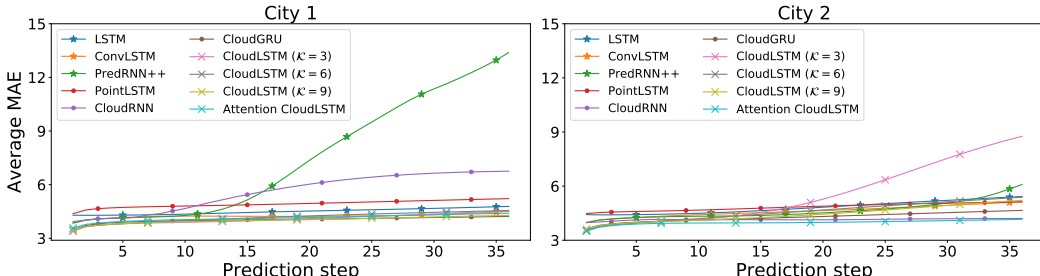

Figure 4: MAE evolution wrt. prediction horizon achieved by RNN-based models on both cities for mobile traffic forecasting.

CloudGRU variants outperform all other banchmark architectures, achieving lower MAE/RMSE and higher PSNR/SSIM on both urban scenarios. This suggests that the DConv operator learns features over geospatial point-clouds more effectively than vanilla convolution and PointCNN. Among our approaches, CloudLSTM performs better than CloudGRU, which in turn outperforms CloudRNN.

Interestingly, the forecasting performance of the CloudLSTM seems fairly insensitive to the number of neighbors ($\mathcal{K}$); it is therefore worth using a small $\mathcal{K}$ in practice, to reduce model complexity. Further, we observe that the attention mechanism improves the forecasting performance, as it helps capturing better dependencies between input sequences and vectors in decoders. This effect has also been confirmed by other NLP tasks. We provide a complete service-wise evaluation in Appendix H.

**Long-term Forecasting Performance.** We extend the prediction horizon to up to $J = 36$ time steps (i.e., 3 hours) for all RNN-based architectures, and show their MAE evolution with respect to this horizon in Fig. 4. Note that the input length remains unchanged, i.e., 6 time steps. In city 1, observe that the MAE does not grow significantly with the prediction step for most models, as the curves flatten. This means that these models are reliable in terms of long-term forecasting. As for city 2, we note that low $\mathcal{K}$ may lead to poorer long term performance for CloudLSTM, though not significant before step 20. This provides a guideline on choosing $\mathcal{K}$ for different forecast lengths required.

## 4.4 RESULTS ON AIR QUALITY FORECASTING

We employ all models to deliver 12-step air quality forecasting on six indicators, given 12 snapshots as input. Results are in Table 2. Also in this use case, the proposed CloudLSTMs attain the best performance across all 4 metrics, outperforming state-of-the-art methods (ConvLSTM) by up to 12.2% and 8.8% in terms of MAE and RMSE, respectively. Unlike in the mobile traffic forecasting results, a lower $\mathcal{K}$ yields better prediction performance, though the difference appears subtle. Again, the CloudCNN always proves superior to the PointCNN, indicating that CloudCNNs are better feature extractors over point-clouds. Overall, these results demonstrate the effectiveness of the CloudLSTM models for modeling spatiotemporal point-cloud stream data, regardless of the tasks to which they are applied. Performance evaluations of long-term forecasting, i.e., up to 72 future time steps, are conducted on RNN-based models and results are presented in Appendix I.

Note that we conduct our experiments using strict variable-controlling methodology, i.e., only changing one factor while keep the remaining the same. Therefore, it is easy to study the effect of each factor. For example, taking a look at the performance of LSTM, ConvLSTM, PredRNN++, PointLSTM and CloudLSTM, which employ dense layers, and CNN, PointCNN and D-Conv as core operators but using LSTM as the RNN structure, it is clear that the D-Conv contributes significantly to the performance improvements. Further, by comparing CloudRNN, CloudGRU and CloudLSTM, it appears that CloudRNN $\ll$ CloudGRU $<$ CloudLSTM. Similarly, by comparing the CloudLSTM and Attention CloudLSTM, we see that the effects of the attention mechanism are not very significant. Therefore, we believe the core operator $>$ RNN structure $>$ attention, ranked by their contribution.

## 5 CONCLUSION

We introduce CloudLSTM, a dedicated neural model for spatiotemporal forecasting tailored to point-cloud data streams. The CloudLSTM builds upon the DConv operator, which performs convolution over point-clouds to learn spatial features while maintaining permutation invariance. The DConv simultaneously predicts the values and coordinates of each point, thereby adapting to changing spatial correlations of the data at each time step. DConv is flexible, as it can be easily combined with various RNN models (i.e., RNN, GRU, and LSTM), Seq2seq learning, and attention mechanisms.

Table 2: The mean±std of MAE, RMSE, PSNR, and SSIM across all models considered, evaluated on two datasets collected in different city clusters for air quality forecasting.

| Model | Cluster A | | | | Cluster B | | | |
|---|---|---|---|---|---|---|---|---|
| | MAE | RMSE | PSNR | SSIM | MAE | RMSE | PSNR | SSIM |
| MLP | 113.13±191.89 | 142.03±240.24 | 23.54±7.38 | 0.13±0.10 | 40.34±22.16 | 50.81±27.27 | 24.75±4.02 | 0.10±0.10 |
| CNN | 37.62±8.18 | 47.67±11.21 | 28.35±2.38 | 0.13±0.05 | 18.59±2.24 | 23.66±2.76 | 30.17±1.14 | 0.34±0.04 |
| 3D-CNN | 37.09±7.63 | 48.01±10.36 | 28.36±2.17 | 0.32±0.08 | 19.84±2.20 | 25.30±2.55 | 29.58±0.99 | 0.35±0.05 |
| DefCNN | 37.51±8.34 | 47.69±11.44 | 28.40±2.41 | 0.13±0.05 | 19.46±2.45 | 25.58±2.80 | 29.55±1.07 | 0.26±0.05 |
| PointCNN | 39.60±7.63 | 51.61±10.35 | 27.63±2.02 | 0.19±0.04 | 19.25±2.38 | 24.60±2.99 | 29.89±1.20 | 0.17±0.03 |
| CloudCNN | 31.62±8.73 | 40.68±11.89 | 29.91±2.89 | 0.23±0.04 | 15.11±3.45 | 19.97±4.33 | 31.91±2.01 | 0.38±0.04 |
| LSTM | 30.62±8.97 | 40.83±11.88 | 29.87±2.79 | 0.31±0.10 | 14.38±3.37 | 19.10±4.29 | 32.16±2.03 | 0.41±0.07 |
| ConvLSTM | 22.91±8.09 | 31.62±11.40 | 31.98±3.24 | 0.50±0.10 | 10.39±2.82 | 14.20±3.87 | 34.79±2.45 | 0.60±0.06 |
| PredRNN++ | 25.14±8.48 | 34.38±11.77 | 31.34±3.13 | 0.37±0.08 | 11.43±2.81 | 15.68±3.85 | 33.94±2.21 | 0.50±0.05 |
| PointLSTM | 36.64±7.99 | 47.42±10.64 | 28.56±2.25 | 0.31±0.06 | 18.77±2.18 | 24.66±2.58 | 29.79±1.07 | 0.35±0.07 |
| CloudRNN ($\mathcal{K}=9$) | 33.09±8.23 | 42.16±11.38 | 29.53±2.66 | 0.13±0.07 | 14.82±3.75 | 19.70±4.54 | 31.93±2.18 | 0.14±0.06 |
| CloudGRU ($\mathcal{K}=9$) | 22.12±8.02 | 30.65±11.25 | 32.22±3.38 | 0.53±0.09 | 9.58±2.82 | 13.41±3.78 | 35.26±2.57 | 0.68±0.07 |
| CloudLSTM ($\mathcal{K}=3$) | **20.84±7.88** | **29.16±11.06** | **32.64±3.40** | **0.57±0.10** | **9.12±2.75** | **12.95±3.72** | **35.59±2.62** | **0.69±0.07** |
| CloudLSTM ($\mathcal{K}=6$) | 21.31±7.52 | 29.71±10.61 | 32.48±3.29 | 0.55±0.10 | 9.38±2.85 | 13.20±2.79 | 35.42±2.60 | 0.68±0.07 |
| CloudLSTM ($\mathcal{K}=9$) | 21.72±7.83 | 30.14±11.05 | 32.36±3.34 | 0.54±0.10 | 9.73±2.84 | 13.58±3.77 | 35.20±2.56 | 0.66±0.07 |
| Attention CloudLSTM ($\mathcal{K}=9$) | 21.72±7.78 | 30.04±10.95 | 32.38±3.29 | 0.56±0.10 | 9.38±2.69 | 13.41±3.78 | 35.26±2.57 | **0.69±0.07** |

We consider two application case studies, where we show that our proposed CloudLSTM achieves state-of-the-art performance on large-scale datasets collected in urban regions in Europe and China. We believe the CloudLSTM gives a new perspective on point-cloud stream modelling, and it can be easily extended to higher dimension point-clouds, without requiring changes to the model.

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

## A DConv Implementation

The DConv can be efficiently implemented using a standard 2D convolution operator, by data shape transformation. We assume a batch size of 1 for simplicity. Recall that the input and output of DConv, $\mathcal{S}'_{in}$ and $\mathcal{S}_{out}$, are 3D tensors with shape $(N, (H+L), U_{in})$ and $(N, (H+L), U_{out})$, respectively. Note that for each $p_n$ in $\mathcal{S}^i_{in}$, we find the set of top $\mathcal{K}$ nearest neighbors $\mathcal{Q}^{\mathcal{K}}_n$. Combining these, we transform the input into a 4D tensor $\mathcal{S}^{i'}_{in}$, with shape $(N, \mathcal{K}, (H+L), U_{in})$. To perform DConv over $\mathcal{S}^{i'}_{in}$, we split the operator into the following steps:

---

**Algorithm 1** Efficient algorithm for DConv implementation using the 2D convolution operator

---

1: **Inputs:**
    $\mathcal{S}'_{in}$, with shape $(N, \mathcal{K}, (H+L), U_{in})$.
2: **Initialise:**
    The weight tensor $\mathcal{W}$.
3: Reshape the input map $\mathcal{S}^{i'}_{in}$ from shape $(N, \mathcal{K}, (H+L), U_{in})$ to shape $(N, \mathcal{K}, (H+L) \times U_{in})$
4: Reshape the weight tensor $\mathcal{W}$ from shape $(U_{in}, \mathcal{K}, (H+L), (H+L), U_{out})$ to shape $(1, \mathcal{K}, U_{in} \times (H+L), U_{out} \times (H+L))$
5: Perform 2D convolution $\mathcal{S}_{out} = Conv(\mathcal{S}^{i'}_{in}, \mathcal{W})$ with step 1 without padding. $\mathcal{S}_{out}$ becomes a 3D tensor with shape $(N, 1, U_{out} \times (H+L))$
6: Reshape the output map $\mathcal{S}_{out}$ to $(N, (H+L), U_{out})$
7: Apply the sigmoid function $\sigma(\cdot)$ to the coordinates feature in $\mathcal{S}_{out}$

---

This enables to translate the DConv into a standard convolution operation, which is highly optimized by existing deep learning frameworks.

## B DConv Complexity Analysis

We study the complexity of DConv by separating the operation into two steps: *(i)* finding the neighboring set $\mathcal{Q}^{\mathcal{K}}_n$ for each point $p_n \in \mathcal{S}$, and *(ii)* performing the weighting computation in Eq. 2. We discuss the complexity of each step separately. For simplicity and without loss of generality, we assume the number of input and output channels are both 1. For step *(i)*, the complexity of finding $\mathcal{K}$ nearest neighbors for one point is close to $O(\mathcal{K} \cdot L \log N)$,[1] if using KD trees (Bentley, 1975). For step *(ii)*, it is easy to see from Eq. 2 that the complexity of computing one feature of the output $p'_n$ is $O((H+L) \cdot \mathcal{K})$. Since each point has $(H+L)$ features and the output point set $\mathcal{S}^j_{out}$ has $N$ points, the overall complexity of step *(ii)* becomes $O(N \cdot \mathcal{K} \cdot (H+L)^2)$. This is equivalent to the complexity of a vanilla convolution operator, where both the input and output have $(H+L)$ channels, and the input map and kernel have $N$ and $\mathcal{K}$ elements, respectively. This implies that, compared to the convolution operator whose inputs, outputs, and filters have the same size, DConv introduces extra complexity by searching the $\mathcal{K}$ nearest neighbors for each point $O(\mathcal{K} \cdot L \log N)$. Such complexity does not increase much even with higher dimensional point clouds.

## C Proof of Transformation Invariance

We show that the normalization of the coordinates features enables transformation invariance with shifting and scaling. The shifting and scaling of a point can be represented as:

$$\varsigma' = A\varsigma + B, \tag{4}$$

---

[1]$L \ll log(n)$ is required to guarantee efficiency. In real life the dimensions of a point cloud dataset are normally 2 or 3, and we usually have significantly more than 3 points in the dataset. Hence, this condition should hold for most applications.

where $A$ and $B$ are a positive scaling coefficient and respectively an offset. By normalizing the coordinates, we have:

$$\begin{aligned}
\varsigma' &= \frac{\varsigma' - \varsigma'_{\min}}{\varsigma'_{\max} - \varsigma'_{\min}} \\
&= \frac{(A\varsigma + B) - (A\varsigma_{\min} + B)}{(A\varsigma_{\max} + B) - (A\varsigma_{\min} + B)} \\
&= \frac{A(\varsigma - \varsigma_{\min})}{A(\varsigma_{\max} - \varsigma_{\min})} \\
&= \frac{\varsigma - \varsigma_{\min}}{\varsigma_{\max} - \varsigma_{\min}}.
\end{aligned} \tag{5}$$

This implies that, by using normalization, the model is invariant to shifting and scaling transformations.

## D  SOFT ATTENTION MECHANISM

We combine our proposed CloudLSTM with the attention mechanism introduced in (Luong et al., 2015). We denote the $j$-th and $i$-th states of the encoder and decoder as $H_{en}^j$ and $H_{de}^i$. The context tensor for state $i$ at the encoder can be represented as:

$$c_i = \sum_{j \in M} a_{i,j} H_{en}^j = \frac{e^{i,j}}{\sum_{j \in M} e_{i,j}}, \tag{6}$$

where $e_{i,j}$ is a score function, which can be selected among many alternatives. In this paper, we choose $e_{i,j} = \mathbf{v}_a^T \tanh(\mathbf{W}_a * [H_{en}^j; H_{de}^i])$. Here $[\cdot; \cdot]$ is the concatenation operator and $*$ is the convolution function. Both $\mathbf{W}_a$ and $\mathbf{v}_a$ are learnable weights. The $H_{de}^i$ and context tensor are concatenated into a new tensor for the following operations.

## E  MODELS CONFIGURATION

We compared our proposal against a set of baselines models. MLP (Goodfellow et al., 2016), CNN (Krizhevsky et al., 2012), and 3D-CNN (Ji et al., 2013) are frequently used as benchmarks in mobile traffic forecasting (Zhang & Patras, 2018; Bega et al., 2019). DefCNN learns the shape of the convolutional filters and has similarities with the DConv operator proposed in this study (Dai et al., 2017). LSTM is an advanced RNN frequently employed for time series forecasting (Hochreiter & Schmidhuber, 1997). While ConvLSTM (Shi et al., 2015) can be viewed as a baseline model for spatiotemporal predictive learning, the PredRNN++ is the state-of-the-art architecture for spatiotemporal forecasting on grid-structural data and achieves the best performance in many applications (Wang et al., 2018). The CloudRNN and CloudGRU have can be formulated as:

**CloudGRU:**

**CloudRNN:**

$$\begin{aligned}
z_t &= \sigma(\mathcal{W}_{sz} \circledast \mathcal{S}_t^v + \mathcal{W}_{hz} \circledast H_{t-1} + b_z), \\
h_t &= \sigma(\mathcal{W}_{sh} \circledast \mathcal{S}_t^v + \mathcal{W}_{sy} \circledast y_{t-1} + b_h), \quad (7) \quad r_t = \sigma(\mathcal{W}_{sr} \circledast \mathcal{S}_t^v + \mathcal{W}_{hr} \circledast H_{t-1} + b_r), \\
y_t &= \sigma(\mathcal{W}_{yh} \circledast h_t + b_y) \qquad\qquad\qquad\qquad H_t' = \tanh(r_t \odot \mathcal{W}_{h'z} \circledast H_{t-1} + \mathcal{W}_{x'z} \circledast \mathcal{S}_t^v) \\
&\qquad\qquad\qquad\qquad\qquad\qquad\qquad\qquad H_t = (1 - z_t) \odot H_t' + z_t \odot H_{t-1}
\end{aligned}$$

$$(8)$$

The CloudRNN and CloudGRU share a similar Seq2seq architecture with CloudLSTM, except that they do not employ the attention mechanism.

We show in Table 3 the detailed configuration along with the number of parameters for each model considered in this study. Note that we used 2 layers (same as our CloudLSTMs) for ConvLSTM, PredRNN++ and PointLSTM, since we found that increasing the number of layers did not improve their performance. $3 \times 3$ filters are commonly used in image applications, where they have been proven effective. This yields a receptive field of 9 ($3 \times 3$), which is equivalent to $\mathcal{K} = 9$ in our CloudLSTMs. Thus this supports a fair comparison. In addition, the PredRNN++ follows a slightly different structure than that of other Seq2seq models, as specified in the original paper.

Table 3: The configuration of all models considered in this study.

| Model | Configuration |
|---|---|
| MLP | Five hidden layers, 500 hidden units for each layer |
| CNN | Eleven 2D convolutional layers, each applies 108 channels and $3 \times 3$ filters, with batch normalization and ReLU functions. |
| 3D-CNN | Eleven 3D convolutional layers, each applies 108 channels and $3 \times 3 \times 3$ filters, with batch normalization and ReLU functions. |
| DefCNN | Eleven 2D convolutional layers, each applies 108 channels and $3 \times 3$ filters, with batch normalization and ReLU functions. Offsets are predicted by separate convolutional layers |
| PointCNN | Eight $\mathcal{X}$-Conv layers, with K, D, P, C=[9, 1, -1, 36] |
| CloudCNN | Eight DConv layers, , with 36 channels and $\mathcal{K} = 9$ |
| LSTM | 2-stack Seq2seq LSTM, with 500 hidden units |
| ConvLSTM | 2-stack Seq2seq ConvLSTM, with 36 channels and $3 \times 3$ filters |
| PredRNN++ | 2-stack Seq2seq PredRNN++, with 36 channels and $3 \times 3$ filters |
| PointLSTM | 2-stack Seq2seq PointLSTM, with K, D, P, C=[9, 1, -1, 36] |
| CloudRNN | 2-stack Seq2seq CloudRNN, with 36 channels and $\mathcal{K} = 9$ |
| CloudGRU | 2-stack Seq2seq CloudGRU, with 36 channels and $\mathcal{K} = 9$ |
| CloudLSTM ($\mathcal{K} = 3$) | 2-stack Seq2seq CloudLSTM, with 36 channels and $\mathcal{K} = 3$ |
| CloudLSTM ($\mathcal{K} = 6$) | 2-stack Seq2seq CloudLSTM, with 36 channels and $\mathcal{K} = 6$ |
| CloudLSTM ($\mathcal{K} = 9$) | 2-stack Seq2seq CloudLSTM, with 36 channels and $\mathcal{K} = 9$ |
| Attention CloudLSTM | 2-stack Seq2seq CloudLSTM, with 36 channels, $\mathcal{K} = 9$ and soft attention mechanism |

## F  LOSS FUNCTION AND PERFORMANCE METRICS

We optimize all architectures using the MSE loss function:

$$\text{MSE}(t) = \frac{1}{|N| \cdot |H|} \sum_{n \in \mathcal{N}} \sum_{h \in \mathcal{H}} ||\widehat{v}_n^h(t) - v_n^h(t)||^2. \tag{9}$$

Here $\widehat{v}_n^h$ is the mobile traffic volume forecast for the $h$-th service, and respectively the forecast value of the $h$-th air quality indicator, at antenna/monitoring station $n$, at time $t$, while $v_n^h$ is the corresponding ground truth. We employ MAE, RMSE, PSNR and SSIM to evaluate the performance of our models. These are defined as:

$$\text{MAE}(t) = \frac{1}{|N| \cdot |H|} \sum_{n \in \mathcal{N}} \sum_{h \in \mathcal{H}} |\widehat{v}_n^h(t) - v_n^h(t)|. \tag{10}$$

$$\text{RMSE}(t) = \sqrt{\frac{1}{|N| \cdot |H|} \sum_{n \in \mathcal{N}} \sum_{h \in \mathcal{H}} ||\widehat{v}_n^h(t) - v_n^h(t)||^2}. \tag{11}$$

$$\text{PSNR}(t) = 20 \log v_{\max}(t) - 10 \log \frac{1}{|N| \cdot |H|} \sum_{n \in \mathcal{N}} \sum_{h \in \mathcal{H}} ||\widehat{v}_n^h(t) - v_n^h(t)||^2. \tag{12}$$

$$\text{SSIM}(t) = \frac{\left(2 \, \widehat{v}_n^h(t) \, \mu_v(t) + c_1\right) \left(2 \, \text{COV}(v_n^h(t), \widehat{v}_n^h(t)) + c_2\right)}{\left(\widehat{v}_n^h(t)^2 \, \mu_v(t)^2 + c_1\right) \left(\text{VAR}(v_n^h(t)) \text{VAR}(\widehat{v}_n^h(t) + c_2\right)}, \tag{13}$$

where $\mu_v(t)$ and $v_{\max}(t)$ are the average and maximum traffic recorded for all services/quality indicators, at all antennas/monitoring stations and time instants of the test set. $\text{VAR}(\cdot)$ and $\text{COV}(\cdot)$ denote the variance and covariance, respectively. Coefficients $c_1$ and $c_2$ are employed to stabilize the fraction in the presence of weak denominators. Following standard practice, we set $c_1 = (k_1 L)^2$ and $c_2 = (k_2 L)^2$, where $L = 2$ is the dynamic range of float type data, and $k_1 = 0.1$, $k_2 = 0.3$.

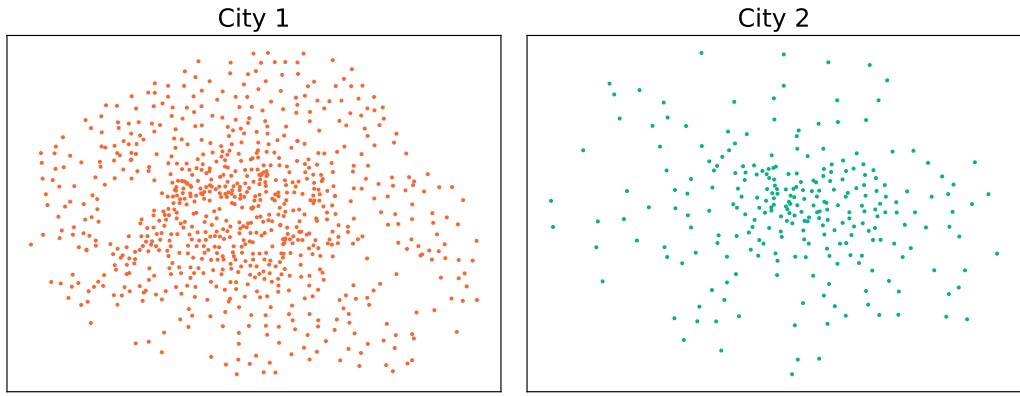

Figure 5: The anonymized locations of the antenna set in both cities.

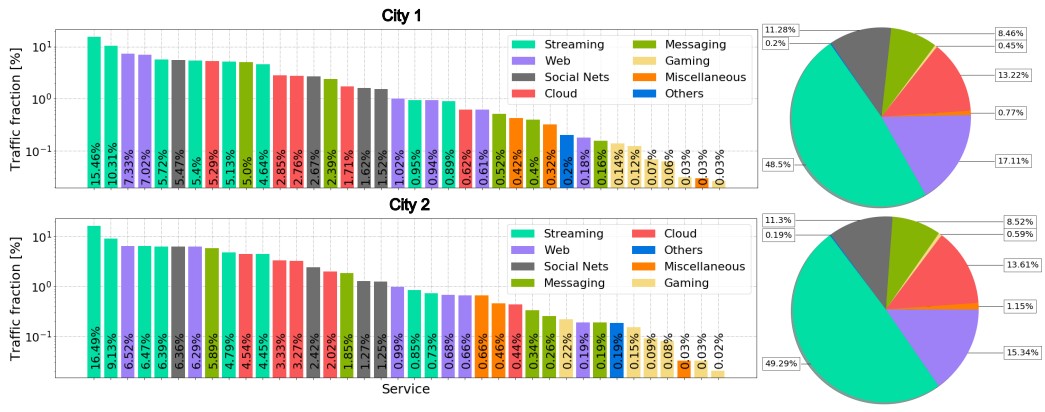

Figure 6: Fraction of the total traffic consumed by each mobile service (left) and each service category (right) in the considered set.

## G DATASET STATISTICS

### G.1 MOBILE TRAFFIC DATASET

#### G.1.1 DATA COLLECTION

The measurement data is collected via traditional flow-level deep packet inspection at the packet gateway (P-GW). Proprietary traffic classifiers are used to associate flows to specific services. Due to data protection and confidentiality constraints, we do not disclose the name of the operator, the target metropolitan regions, or the detailed operation of the classifiers. For similar reasons, we cannot name the exact mobile services studied. We show the anonymized locations of the antennas sets in both cities in Fig. 5

As a final remark on data collection, we stress that all measurements were carried out under the supervision of the competent national privacy agency and in compliance with applicable regulations. In addition, the dataset we employ for our study only provides mobile service traffic information accumulated at the antenna level, and does not contain personal information about individual subscribers. This implies that the dataset is fully anonymized and its use for our purposes does not raise privacy concerns. Due to a confidentiality agreement with the mobile traffic data owner, the raw data cannot be made public.

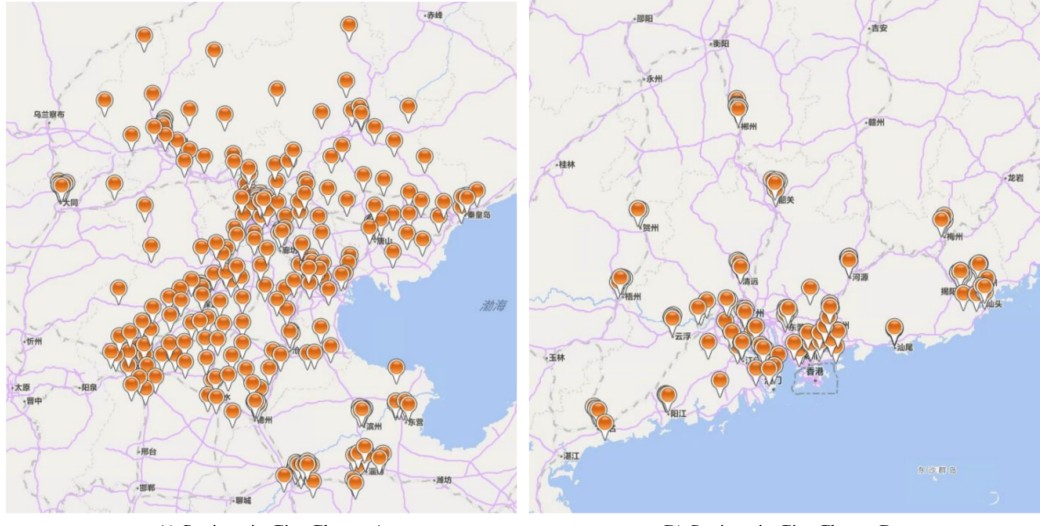

| A) Stations in City Cluster A | B) Stations in City Cluster B |

Figure 7: Geographic distribution of 437 air quality monitoring stations in both city clusters. Figures adapted from the readme.pdf file included with the dataset.

### G.1.2 SERVICE USAGE OVERVIEW

As already mentioned, the set of services $\mathcal{S}$ considered in our analysis comprises 38 different services. An overview of the fraction of the total traffic consumed by each service and each category in both cities throughout the duration of the measurement campaign is in Fig. 6. The left plot confirms the power law previously observed in the demands generated by individual mobile services. Also, streaming is the dominant type of traffic, with five services ranking among the top ten. This is confirmed in the right plot, where streaming accounts for almost half of the total traffic consumption. Web, cloud, social media, and chat services also consume large fractions of the total mobile traffic, between 8% and 17%, whereas gaming only accounts for 0.5% of the demand.

### G.2 AIR QUALITY DATASET

The air quality dataset comprises air quality information from 43 cities in China, collected by the Urban Computing Team at Microsoft Research. In total, there are 2,891,393 air quality records from 437 air quality monitoring stations, gathered over a period of one year. The stations are partitioned into two clusters, based on their geographic locations, as shown in Fig. 7. Cluster A has 274 stations, while Cluster B includes 163. Note that missing data exists in the records and gaps have been filled through linear interpolation. The dataset is available at https://www.microsoft.com/en-us/research/project/urban-air/.

## H  SERVICE-WISE EVALUATION

We dive deeper into the performance of the proposed Attention CloudLSTMs, by evaluating the forecasting accuracy for each individual mobile service, averaged over 36 steps. To this end, we present the MAE evaluation on a service basis (left) and category basis (right) in Fig. 8. Observe that the attention CloudLSTMs obtain similar performance over both cities at the service and category level. Jointly analyzing with Fig. 6, we see that services with higher traffic volume on average (e.g., streaming and cloud) also yield higher prediction errors. This is because their traffic evolution exhibits more frequent fluctuations, which introduces higher uncertainty, making the traffic series more difficult to predict.

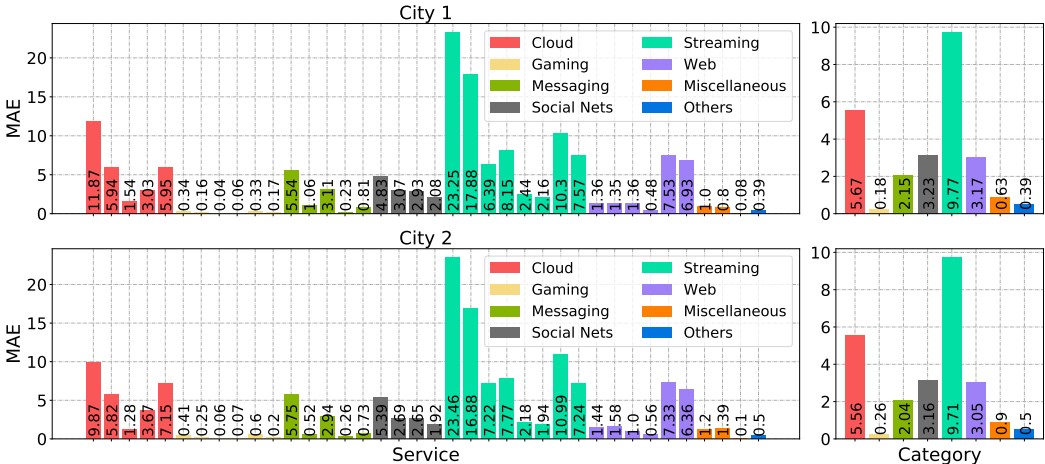

Figure 8: Mobile service-level MAE evaluation on both cities for the Attention CloudLSTMs, averaged over 36 prediction steps.

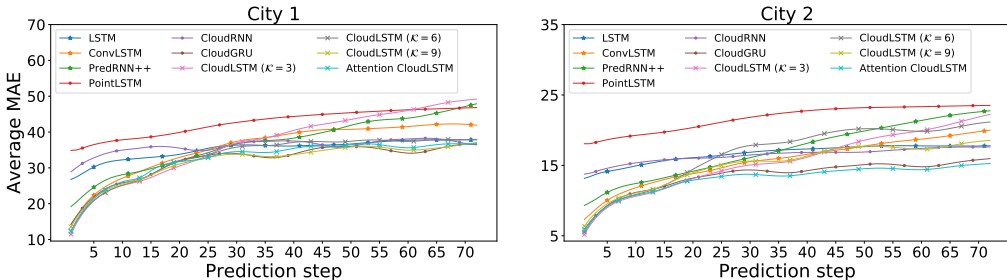

Figure 9: MAE evolution wrt. prediction horizon achieved by RNN-based models on both city clusters for the air quality forecasting.

## I  LONG-TERM AIR QUALITY FORECASTING

We show the MAE for long-term forecasting (72 steps) of air quality on both city clusters in Fig. 9. Generally, the error grows with time for all models, as expected. Turning attention to the CloudLSTM with different $\mathcal{K}$, though the performance of different settings appears similar at the beginning, larger $\mathcal{K}$ can significantly improve the robustness of the CloudLSTM, as the MAE grows much slower with time when $\mathcal{K} = 9$. This is consistent with the conclusion made in the mobile traffic forecasting task.

## J  RESULTS VISUALIZATION

### J.1  HIDDEN FEATURE VISUALIZATION

We complete the evaluation of the mobile traffic forecasting task by visualizing the hidden features of the CloudLSTM, which provide insights into the knowledge learned by the model. In Fig. 10, we show an example of the scatter distributions of the hidden state in $H_t$ of CloudLSTM and Attention CloudLSTM at both stacks, along with the first input snapshots. The first 6 columns show the $H_t$ for encoders, while the rest are for decoders. The input data snapshots are samples selected from City 2 (260 antennas/points). Recall that each $H_t$ has 1 value features and 2 coordinate features for each point, therefore each scatter subplot in Fig. 10 shows the value features (volume represented by different colors) and coordinate features (different locations), averaged over all channels. Observe that in most subplots, points with higher values (warm colors) tend to aggregate into clusters and have higher densities. These clusters exhibit gradual changes from higher to lower values, leading to comet-shape assemblages. This implies that points with high values also come with tighter spatial correlations, thus CloudLSTMs learn to aggregate them. This pattern becomes more obvious in

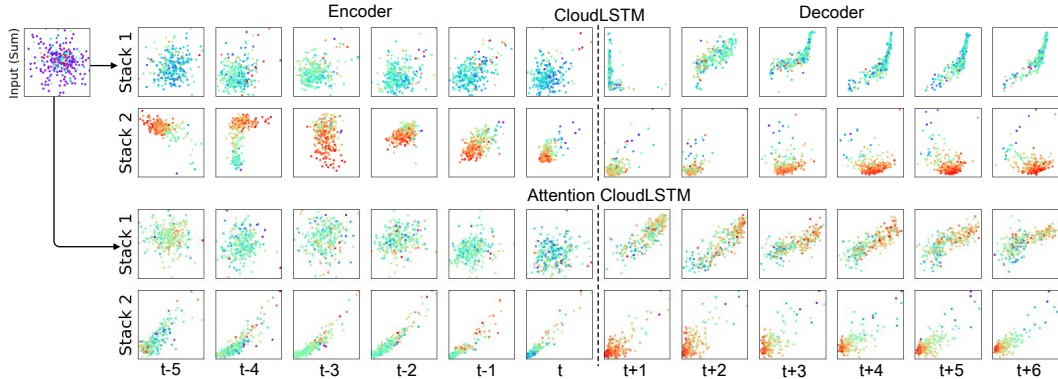

Figure 10: The scatter distributions of the value and coordinate features of the hidden state in $H_t$ for CloudLSTM and Attention CloudLSTM. Values and coordinates are averaged over all channels.

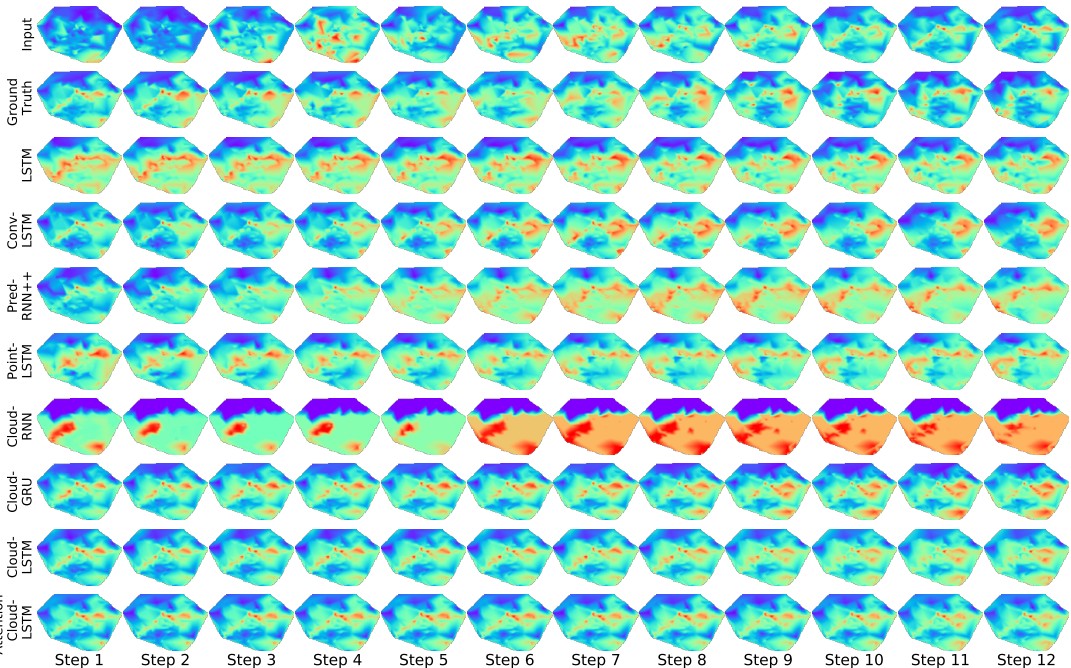

Figure 11: NO$_2$ forecasting examples in City Cluster A generated by all RNN-based models.

stack 2, as features are extracted at a higher level, exhibiting more direct spatial correlations with respect to the output.

### J.2 PREDICTION RESULTS VISUALIZATION

Lastly, in Fig. 11 and 12 and we show a set of NO$_2$ forecasting examples in both city cluster A and B considered for air quality prediction, generated by all RNN-based models, offering a performance comparison from a purely visual perspective. Point-clouds are converted into heat maps using 2D linear interpolation. The better prediction offered by (Attention) CloudLSTMs is apparent, as our proposed architectures capture trends in the point-cloud streams and deliver high long-term visual fidelity, whereas the performance of other architectures degrade rapidly in time.

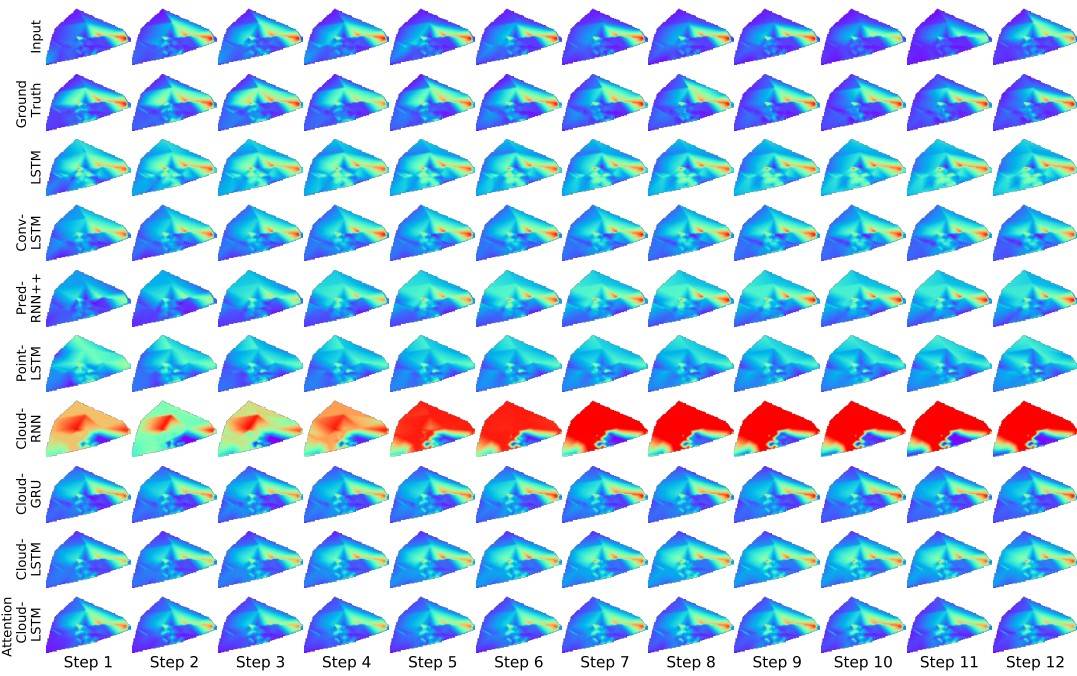

Figure 12: NO$_2$ forecasting examples in Cluster B generated by the RNN-based models.

## K ROBUSTNESS TO OUTLIERS

The DConv uses Sigmoid functions to regularize the coordinate features of each point, such that those points which are far from others will move closer to each other and be more involved in the computation. Further, by stacking multiple DConv via dedicated structure (LSTM), the CloudLSTM has much stronger representability and therefore allows to refine the positions of each input point at each time step and each stack. Eventually, each point can learn to move to the position where it is best to be and therefore, our model continues to work well while forecasting with outlier points.

For demonstration, we use the density-based spatial clustering of applications with noise (DBSCAN) to find those outliers (red points) in both clusters in the air quality dataset, as shown in Fig. 13. For each city cluster, the DBSCAN algorithm finds 16 outlier points, which are relatively isolated and far from the point-cloud center. We recompute the MAE and RMSE performance especially for these outlier points, as shown in Table 4. Observe that our CloudLSTM still obtains the lowest prediction error as compared to the other models considered. Taking a closer look at the CNN-based models, the CloudCNN, which employs the DConv operator, obtains the best forecasting performance relative to CNN, 3D-CNN, DefCNN and PointCNN.

We dive deeper into the robustness of our CloudLSTM to outliers by conducting experiments under more controlled scenarios. To this end, we randomly selected 50 weather stations in each city cluster and construct a toy dataset. Among these weather stations, we randomly pick 10 as outliers, and move their positions away from the center by $d = \{0, 0.5, 1, 5\}$ on both x and y axes. The direction of movement depends on the quadrant of each outlier. Note that the original position of each weather station is normalized to $[0, 1]$, so $d = 5$ means the point is moved at a distance 5 times the maximum range of its original position. The positions of the remaining 40 weather stations remain unchanged. We show the positions of each weather stations after moving by different $d$ for both city clusters in Fig. 14 and 15.

We retrain the CloudLSTM and PointLSTM under the same settings, and show the MAE and RMSE performance of each in Table. 5. Observe that the proposed CloudLSTM performs almost equally well when forecasting over inliers and outliers, irrespective of the distance to outliers. Importantly, CloudLSTM achieves significantly better performance over its counterpart PointLSTM. This further demonstrates that our model is robust to outliers, whose locations appear "lone".

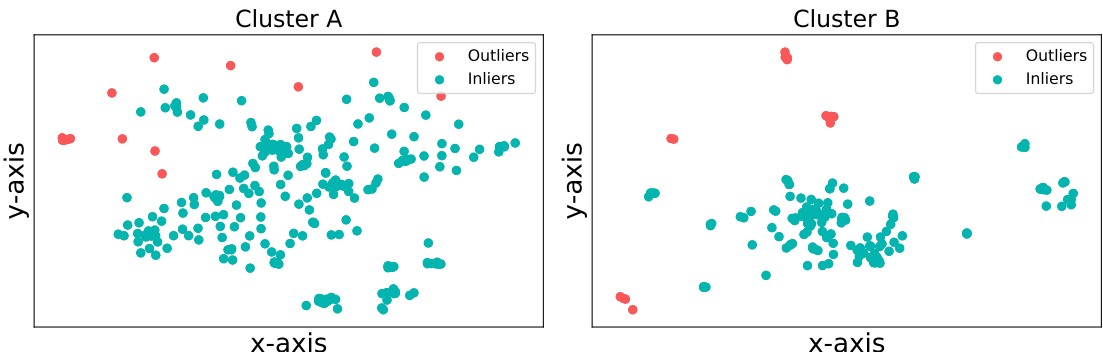

Figure 13: Clustering results in the two city clusters using DBSCAN.

Table 4: The mean±std of MAE and RMSE across all models considered, evaluated on outlier points in different city clusters for air quality forecasting.

| Model | Cluster A | | Cluster B | |
|---|---|---|---|---|
| | **MAE** | **RMSE** | **MAE** | **RMSE** |
| MLP | 109.85±195.49 | 136.36±240.77 | 45.59±20.82 | 56.72±25.31 |
| CNN | 33.04±10.51 | 42.63±14.49 | 21.67±2.88 | 26.53±3.65 |
| 3D-CNN | 32.36±10.48 | 41.49±14.73 | 24.91±3.97 | 30.82±4.91 |
| DefCNN | 33.38±10.13 | 43.0±14.18 | 25.72±3.34 | 33.06±4.26 |
| PointCNN | 34.71±10.79 | 44.56±14.65 | 24.56±2.88 | 30.79±3.49 |
| CloudCNN | 31.79±10.72 | 40.52±13.63 | 15.54±3.54 | 19.26±4.91 |
| LSTM | 27.15±10.08 | 35.85±14.31 | 15.62±4.13 | 19.72±5.5 |
| ConvLSTM | 21.57±9.79 | 28.94±14.16 | 11.18±3.06 | 14.5±4.38 |
| PredRNN++ | 24.34±9.82 | 32.78±13.95 | 12.53±3.55 | 16.43±4.98 |
| PointLSTM | 32.2±11.01 | 41.39±15.25 | 21.28±4.16 | 26.63±5.38 |
| CloudRNN ($\mathcal{K} = 9$) | 32.34±8.57 | 40.17±12.6 | 15.01±3.98 | 18.60±5.23 |
| CloudGRU ($\mathcal{K} = 9$) | 20.52±9.67 | 27.49±13.86 | 9.59±3.01 | 12.64±4.32 |
| CloudLSTM ($\mathcal{K} = 3$) | **18.79±9.44** | **25.59±13.76** | **9.04±3.14** | **11.99±4.45** |
| CloudLSTM ($\mathcal{K} = 6$) | 20.3±8.99 | 27.23±13.12 | 9.54±3.25 | 12.55±4.54 |
| CloudLSTM ($\mathcal{K} = 9$) | 20.09±9.53 | 27.24±13.76 | 9.77±3.35 | 12.79±4.68 |
| Attention CloudLSTM ($\mathcal{K} = 9$) | 19.79±9.48 | 26.63±13.66 | 9.5±3.14 | 12.49±4.48 |

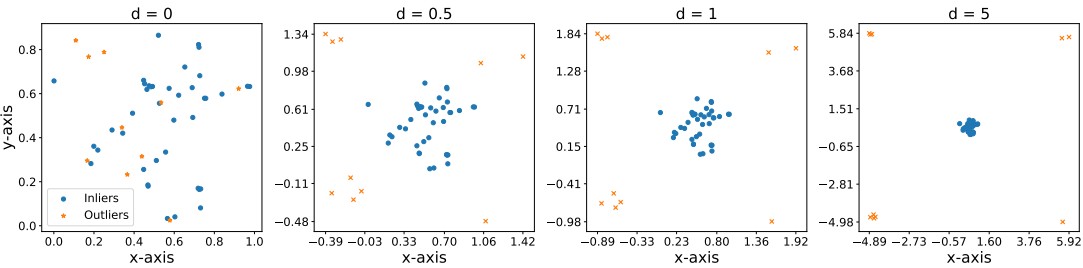

Figure 14: Positions of weather stations after moving outliers towards the edge with different distances $d$ in city cluster A.

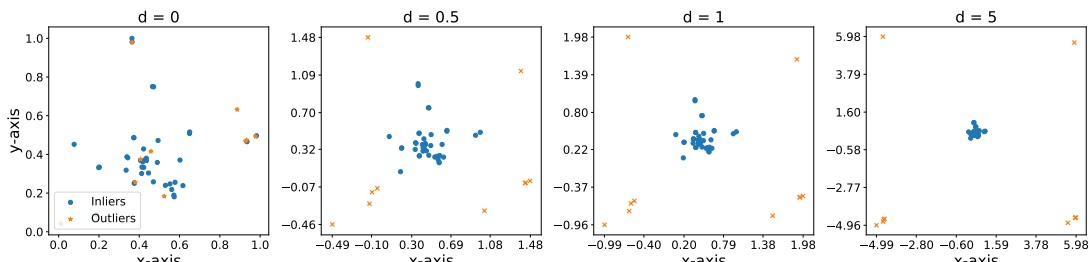

Figure 15: Positions of weather stations after moving outliers towards the edge with different distances $d$ in city cluster B.

Table 5: The mean±std of MAE and RMSE across CloudLSTM ($\mathcal{K} = 9$) and PointLSTM, evaluated on inliers and outliers of two sub-datasets collected in different city clusters for air quality forecasting.

| Model | $d$ | Cluster A | | | | Cluster B | | | |
|---|---|---|---|---|---|---|---|---|---|
| | | MAE | | RMSE | | MAE | | RMSE | |
| | | Inliers | Outliers | Inliers | Outliers | Inliers | Outliers | Inliers | Outliers |
| CloudLSTM | 0 | 22.16±7.28 | 20.39±9.61 | 29.53±9.39 | 28.55±8.76 | 9.78±5.63 | 9.06±4.11 | 13.10±3.76 | 13.28±4.56 |
| | 0.5 | 22.75±7.29 | 20.5±8.66 | 29.16±9.44 | 28.80±7.25 | 10.07±5.50 | 9.35±4.75 | 13.49±3.63 | 13.59±4.13 |
| | 1 | 22.4±7.30 | 19.98±9.53 | 29.04±9.58 | 28.41±8.36 | 10.03±5.40 | 9.21±4.53 | 13.33±3.26 | 13.56±4.92 |
| | 5 | 21.98±7.31 | 19.5±9.78 | 28.63±9.52 | 28.2±8.73 | 9.34±5.52 | 9.64±4.66 | 12.51±3.63 | 12.69±4.06 |
| PointLSTM | 0 | 40.72±20.01 | 40.06±18.35 | 58.68±34.19 | 58.30±31.69 | 20.08±10.39 | 18.72±8.33 | 28.38±14.67 | 27.12±13.22 |
| | 0.5 | 38.45±17.17 | 39.18±16.64 | 55.32±31.25 | 57.48±30.06 | 19.97±9.53 | 19.49±6.98 | 28.16±13.69 | 27.88±10.84 |
| | 1 | 38.08±17.5 | 36.51±17.05 | 54.85±31.74 | 54.1±31.48 | 19.29±9.65 | 18.40±7.35 | 27.38±13.86 | 26.84±12.1 |
| | 5 | 36.4±16.70 | 39.13±16.93 | 52.55±30.82 | 56.81±29.82 | 17.49±7.56 | 20.34±9.34 | 25.05±11.03 | 29.35±13.67 |

## L  ADDITIONAL EVALUATION WITH SIMPLE BASELINES

We further compare our proposal with simple baselines, which also perform forecasting based on k-nearest neighbors of each target point. To this end, we construct MLPs and LSTMs with the structures specified in Table 3, but with different input form. Specifically, for each point, the models perform prediction using only the $\mathcal{K}$ nearest neighbors' data, with $\mathcal{K}$ from $\{1, 3, 6, 9, 25, 50, 100\}$. We show their performance along with that of our CloudLSTM on the air quality dataset in Table 6. Observe that our CloudLSTM significantly outperforms MLPs and LSTMs, which conduct forecasting only relying on k-nearest neighbors. The number of neighbors $\mathcal{K}$ affects the receptive field of each model. A small $\mathcal{K}$ means the model only relies on limited local spatial dependencies, while global spatial correlations between points are neglected. In contrast, a large $\mathcal{K}$ enables looking around larger location spaces, while this might lead to overfitting. The results in the table suggest that the $\mathcal{K}$ does not affect the performance of each baseline significantly. Meanwhile our proposed CloudLSTM, which extracts local spatial dependencies through DConv kernels and merges global spatial dependency via stacks of time steps and layers, is superior to these simple baselines.

Table 6: The mean±std of MAE, RMSE, PSNR, and SSIM across k-nearest neighbors baseline models and our CloudLSTM, evaluated on two datasets collected in different city clusters for air quality forecasting.

| Model | Cluster A | | | | Cluster B | | | |
|---|---|---|---|---|---|---|---|---|
| | MAE | RMSE | PSNR | SSIM | MAE | RMSE | PSNR | SSIM |
| MLP ($\mathcal{K} = 1$) | 31.28±7.29 | 38.72±9.68 | 29.95±2.28 | 0.48±0.11 | 14.33±2.06 | 17.77±2.78 | 32.20±1.33 | 0.61±0.06 |
| MLP ($\mathcal{K} = 3$) | 29.12±7.33 | 38.36±9.42 | 30.17±2.26 | 0.46±0.10 | 15.35±2.12 | 19.20±2.28 | 31.16±1.11 | 0.59±0.06 |
| MLP ($\mathcal{K} = 6$) | 29.53±7.24 | 38.35±10.01 | 30.06±2.52 | 0.45±0.12 | 16.73±2.22 | 20.99±2.31 | 30.66±1.01 | 0.58±0.05 |
| MLP ($\mathcal{K} = 9$) | 29.73±7.94 | 38.62±11.12 | 30.15±2.72 | 0.40±0.10 | 18.73±2.40 | 21.98±2.64 | 30.23±1.09 | 0.58±0.06 |
| MLP ($\mathcal{K} = 25$) | 31.59±7.63 | 40.34±11.39 | 29.82±2.54 | 0.30±0.09 | 18.93±2.47 | 22.98±3.25 | 30.37±1.31 | 0.22±0.04 |
| MLP ($\mathcal{K} = 50$) | 30.5±8.02 | 39.02±10.72 | 30.07±2.60 | 0.29±0.12 | 14.62±2.44 | 18.3±3.37 | 32.39±1.55 | 0.54±0.07 |
| MLP ($\mathcal{K} = 100$) | 30.39±8.55 | 38.99±11.53 | 30.1±2.78 | 0.26±0.13 | 14.27±3.32 | 18.65±4.13 | 32.32±1.95 | 0.39±0.05 |
| LSTM ($\mathcal{K} = 1$) | 24.31±7.52 | 32.52±10.65 | 31.67±3.02 | 0.53±0.10 | 11.47±2.91 | 15.50±3.89 | 34.51±2.45 | 0.67±0.07 |
| LSTM ($\mathcal{K} = 3$) | 24.33±7.51 | 32.46±10.57 | 31.54±3.13 | 0.53±0.10 | 11.46±2.94 | 15.45±3.96 | 34.55±3.44 | 0.65±0.08 |
| LSTM ($\mathcal{K} = 6$) | 24.66±7.77 | 32.97±10.61 | 31.49±2.79 | 0.53±0.10 | 11.52±2.89 | 15.46±4.13 | 34.50±2.47 | 0.64±0.07 |
| LSTM ($\mathcal{K} = 9$) | 24.89±7.68 | 33.07±10.64 | 31.55±2.98 | 0.52±0.10 | 11.59±3.00 | 15.77±4.09 | 34.40±2.52 | 0.62±0.07 |
| LSTM ($\mathcal{K} = 25$) | 25.46±7.91 | 33.8±10.99 | 31.33±3.04 | 0.50±0.10 | 11.58±3.06 | 15.74±4.12 | 34.47±2.56 | 0.60±0.06 |
| LSTM ($\mathcal{K} = 50$) | 25.68±7.87 | 33.98±10.91 | 31.27±2.97 | 0.48±0.10 | 11.72±3.26 | 15.81±4.28 | 34.48±2.67 | 0.59±0.06 |
| LSTM ($\mathcal{K} = 100$) | 25.57±8.73 | 34.53±11.87 | 31.17±3.17 | 0.44±0.09 | 11.37±2.82 | 15.47±3.76 | 33.97±2.22 | 0.55±0.05 |
| CloudLSTM ($\mathcal{K} = 3$) | **20.84±7.88** | **29.16±11.06** | **32.64±3.40** | **0.57±0.10** | **9.12±2.75** | **12.95±3.72** | **35.59±2.62** | **0.69±0.07** |

Table 7: The mean±std of MAE and RMSE across all models considered without/with seasonal information, evaluated on a subset of antennas in City 1 for mobile traffic forecasting.

| Model | 30 Minutes Window | | 30 Minutes + 7 Days Window | |
|---|---|---|---|---|
| | **MAE** | **RMSE** | **MAE** | **RMSE** |
| MLP | 4.86±0.51 | 10.30±2.52 | 4.93±0.53 | 11.30±2.2 |
| CNN | 6.10±0.59 | 11.12±2.04 | 5.98±0.60 | 10.52±2.11 |
| 3D-CNN | 5.01±0.51 | 9.89±2.54 | 4.82±0.54 | 9.49±2.36 |
| DefCNN | 6.79±0.89 | 11.92±2.47 | 6.40±0.83 | 11.55±2.33 |
| PointCNN | 5.01±0.55 | 10.22±2.40 | 4.88±0.51 | 9.86±2.31 |
| CloudCNN | 4.79±0.53 | 9.94±2.75 | 4.63±0.50 | 9.57±2.66 |
| LSTM | 4.24±0.64 | 9.67±3.23 | 4.04±0.67 | 9.28±3.03 |
| ConvLSTM | 4.10±1.61 | 9.28±3.11 | 3.82±1.54 | 8.87±3.00 |
| PredRNN++ | 3.94±1.62 | 9.31±3.10 | 3.61±1.55 | 8.95±2.92 |
| PointLSTM | 4.63±0.41 | 9.44±2.46 | 4.44±0.41 | 9.00±2.32 |
| CloudRNN ($\mathcal{K} = 9$) | 4.14±1.67 | 9.18±3.13 | 3.99±1.62 | 8.88±2.93 |
| CloudGRU ($\mathcal{K} = 9$) | 3.77±1.58 | 8.95±3.08 | 3.42±1.53 | 8.53±2.88 |
| CloudLSTM ($\mathcal{K} = 3$) | 3.69±1.61 | 8.83±3.09 | 3.38±1.57 | 8.40±2.86 |
| CloudLSTM ($\mathcal{K} = 6$) | 3.68±1.60 | 8.87±3.10 | 3.39±1.54 | 8.43±2.76 |
| CloudLSTM ($\mathcal{K} = 9$) | 3.69±1.61 | 8.86±3.12 | 3.40±1.56 | 8.46±2.77 |
| Attention CloudLSTM ($\mathcal{K} = 9$) | **3.60±1.59** | **8.77±3.06** | **3.22±1.55** | **8.36±2.66** |

## M   AUGMENTING FORECASTING WITH SEASONAL INFORMATION

Finally, we notice that seasonal information exists in the mobile traffic series, which can be further exploited to improve the forecasting performance. However, directly feeding the model with data spanning multiple days is infeasible, since, e.g., a 7-day window corresponds to a 2016-long sequence as input (given that data is sampled every 5 minutes) and it is very difficult for RNN-based models to handle such long sequences. In addition, by considering the number of mobile services (38) and antennas (792), the input for 7 days would have 60,673,536 data points. This would make any forecasting model extremely large and therefore impractical for real deployment.

To capture seasonal information more efficiently, we concatenate the 30 minute-long sequences (sampled every 5 minutes) with a sub-sampled 7-day window (sampled every 2h). This forms an input with length 90 (6 + 84). We conduct experiments on a randomly selected subset (100 antennas) of the mobile traffic dataset (City 1), and show the forecasting performance without and with seasonal information (7-day window) in Table 7. By incorporating the seasonal information, the performance of most forecasting models is boosted. This indicates that the periodic information is learnt by the model, which helps reduce the prediction errors. However, the concatenation increases the length of the input, which also increases the model complexity. Future work will focus on a more efficient way to fuse the seasonal information, with marginal increase in complexity.

