# OpenReview forum: "CloudLSTM: A Recurrent Neural Model for Spatiotemporal Point-cloud Stream Forecasting"
_ICLR.cc/2020/Conference — Reject_

### Official Review · AnonReviewer1 · 2019-10-22
**Official Blind Review #1**

**Rating:** 3

**Review:**

Review Summary
--------------
Overall this is almost above the bar for me to accept, but I think there's enough concerns about the method and experiments that I'm hesitant. Strengths include the invariance to point cloud order and the relative simplicity of the architecture (compared to PointCNN). Weaknesses include a vulnerability to outliers, experiments that don't seem to think about practical effects like day-of-week in forecasting, and experiments that leave out baselines to help directly assess the impact of neighbors.


Paper Summary
-------------
The paper develops a new neural net architecture for processing data structured as spatial point clouds that vary over time (e.g. hourly traffic at several antennas spread throughout a city).

The core of the approach is a new neural net unit: the "D-Conv" operator (See Eq. 2). The output value at each point is obtained via a weighted combination of nearby coordinates and features, using only the K-nearest neighbors (stored in ranked order) to maintain invariance to the original order of points. This layer can be included in modern convolutional (CloudCNN) or recurrent (CloudLSTM) or attention-based architectures in a straightforward way.

Unlike many previous methods that require converting point clouds to quantized regular grids, the present approach directly consumes point cloud data. Unlike some existing methods like PointCNN, it avoids information loss (does not reduce dimension from input to output layer).

Two experimental evaluations are conducted: forecasting mobile app traffic across 2 European cities (given past 30 min, predict next 30 min), and air quality across several regions in China (given last 12 hrs, predict next 12 hrs). In both experiments, the locations of the sensors are fixed across time. Fig 4 further looks at traffic forecasting as a function of the lookahead time, from 0-3 hours ahead.


Novelty & Significance
-----------------------

The paper definitely tackles an important problem (point cloud forecasting).

The present paper's new "D-Conv" operator appears new, though it looks really like a simplification of the PointCNN's "X-Conv" operator rather than a brand new operator.

The most similar work seems to be the PointCNN (Li et al NeurIPS 2018). This work's contribution was a new "X-Conv" operator, which also consumes point clouds and produces learned representations. X-Conv, like the present paper's D-Conv, computes K-nearest neighbors of each point p, but performs first an embedding of each neighbor to a learned "local" feature space and then performs convolution on this embedding. Perhaps the biggest practical difference is that D-Conv has fewer parameters (does not perform the embedding) and does not reduce dimensionality from input to output.

Technical Concerns
------------------

My biggest concerns are that the D-Conv has a strong reliance on nearest neighbors. This means the D-Conv has not much accomodation for "outlier" points that are far from others. The X-Conv operator has some nice properties in this regard (it changes coordinate systems so neighbor locations are centered around the current point), but I don't see this in the D-Conv operator, as in Eq. 2, where the coordinate locations are fed directly into the weighted sum after global rescaling to (0,1). I would imagine that data with outliers (whose values are unlike most others) would dramatically hurt performance, as the weights of D-Conv would need to be shared equally by outliers and inliers.


Experimental Concerns
---------------------

Is there a good reason to not try to compare on publicly available datasets like those used in the PointCNN paper (focusing only on the non-temporal versions of the model)? Using proprietary datasets makes following up on this work a bit hard, would be nice to have some reproducible experiment.

It's not clear to me that the experiments here consider realistic scenarios. Why would I predict mobile app traffic using only the past 30 minutes of data? Why predict air quality using only the last 12 hours? Certainly there are time-of-day, day-of-week, and seasonal effects that are all important. At a minimum, I'd think that for the mobile traffic case you could at least look at consuming the last 48 hr of data and predicting the next 30-90 minutes. I suspect that would make even simpler models do much better.

Further, I think the experiments are missing some key simple baselines (or I misunderstand something). For example, rather than the complicated CNN/LSTM architectures, why not try to directly see how much value there is in "neighbors" in this 2d space? At each point, you can make predictions using only the K nearest neighbors' data, with K swept from 1 to 100 or something. I would expect with these features, using just a simple MLP or RNN would do quite well. I'd like to see a stronger qualitative case made for why we expect the complicated DConv weighting operator here to do better than this baselines.

Overall, the results tables appear promising (for app traffic forecasting in Table 1, the proposed CloudLSTM achieves 3.66 MAE compared to 4.95 for PointCNN and 4.8 for an MLP). However, it's not clear why and I'd like to understand why. Is it that the other approaches are overfitting?


Minor Concerns
--------------
I would suggest avoiding calling the method "\mathcal{D}-Conv", and instead use just "DConv", since this is easier to type into search engines and easier to search for in a PDF document

Related: Point clouds could be represented as graphs, and then use graph embeddings as feature representations

**Experience Assessment:**

I have read many papers in this area.

**Review Assessment: Checking Correctness Of Derivations And Theory:**

I did not assess the derivations or theory.

**Review Assessment: Checking Correctness Of Experiments:**

I carefully checked the experiments.

**Review Assessment: Thoroughness In Paper Reading:**

I read the paper at least twice and used my best judgement in assessing the paper.

---

> ### Author Response · Authors · 2019-11-10
> **Response to Blind Review #1 (1)**
>
> Thank you for acknowledging the contribution of our paper and offering insightful comments. We address the concerns raised below.
>
> Q: The present paper's new D-Conv operator appears new, though it looks really like a simplification of the PointCNN's "X-Conv" operator rather than a brand new operator. The most similar work seems to be the PointCNN (Li et al NeurIPS 2018). This work's contribution was a new X-Conv operator, which also consumes point-clouds and produces learned representations. X-Conv, like the present paper's D-Conv, computes K-nearest neighbors of each point p, but performs first an embedding of each neighbor to a learned "local" feature space and then performs convolution on this embedding. Perhaps the biggest practical difference is that D-Conv has fewer parameters (does not perform the embedding) and does not reduce dimensionality from input to output.
>
> A: Our D-Conv is inspired by convolution over grids, which computes summations over small regions, as we also do but over point-clouds. This may of course introduce some problems, such as the permutation of the points. X-Conv addresses this by an X-transformation, while our work solves the problem by keeping the order of the k-nearest-neighbor weights. Therefore, we believe this is a different way of solving the problem, rather than a simplification of the X-Conv. Further, different from X-Conv, our D-Conv treats the value and coordinate features differently, by regularizing the coordinate features using a sigmoid function to avoid outliers in space. By looking at the performance of PointLSTM (X-Conv + LSTM, which is also a new model we derive) and  CloudLSTM (D-Conv + LSTM), our CloudLSTM performs much better, which proves that D-Conv is more suitable for the forecasting task.
>
> Importantly, we want to note that the D-Conv is not the only contribution of our paper. Combining a convolutional operator dedicated for point-cloud with an RNN structure for forecasting is also an important contribution and, to the best of our knowledge, we are the first to do this.
>
>
> Q: My biggest concerns are that the D-Conv has a strong reliance on nearest neighbors. This means the D-Conv has not much accommodation for "outlier" points that are far from others.  I would imagine that data with outliers (whose values are unlike most others) would dramatically hurt performance, as the weights of D-Conv would need to be shared equally by outliers and inliers.
>
> A: In Eq 2, we regularize the coordinate features of each point using a Sigmoid function, such that those points which are far from others will move closer to each other and will be more involved in the computation. This is exactly how the D-Conv handles outliers. Further, though the weights are shared by outliers and inliers, the final forecasts are made by multiple stacks of CloudLSTM and through multiple steps of computation. This means that the model would have sufficient representability to move those outliers to positions where they are best to be, thus the performance will not be compromised.
>
> For demonstration, we add a new section in Appendix K in the revised paper, to test the performance of each model especially with outlier points found by DBSCAN in the air quality dataset. Observe that our CloudLSTM remains the best when performing forecasting with outliers. Compared with the full forecasting in Table 2, our proposals achieve even better performance, which proves that the CloudLSTM remains reliable when forecasting over outliers.
>
> Q: Is there a good reason to not try to compare on publicly available datasets like those used in the PointCNN paper (focusing only on the non-temporal versions of the model)? Using proprietary datasets makes following up on this work a bit hard, would be nice to have some reproducible experiment.
>
> A: The datasets in the PointCNN paper are dedicated to point-cloud classification and segmentation, while our work focuses on point-cloud stream forecasting, hence the scope is different. Since the D-Conv, along with the CloudLSTM, are designed for the task of temporal forecasting, we do not test the model on those datasets.
>
> We fully agree however that it would be good to have a reproducible experiment with publicly available datasets. While the mobile traffic dataset can not be released, the air quality datasets are publicly available and can be found at https://www.microsoft.com/en-us/research/project/urban-air/.
> We will release the code and processed dataset upon final decision, so as to support the reproducibility of our results.
>
> to be continued...

---

> > ### Comment · AnonReviewer1 · 2019-11-13
> > **Thanks for a thorough rebuttal; some concerns remain**
> >
> > Overall, I'm still a bit borderline on this paper, but I'm leaning more positive than before based on the thorough rebuttal. If others are voting for acceptance I won't stand strongly against it. I'm glad authors are thinking about a reproducible version of the urban-air dataset and were willing to implement a few baselines I suggested.
> >
> > RE experiments not considering day-of-week / seasonality effects
> >
> > I am not quite satisfied with this explanation. I find it a bit hard to believe that storing (sparser) traffic data long enough to capture day of week effects wouldn't be worth it. Surely there is more Saturday activity in downloading game apps than weekday activity, why throw this away? Plus, I think the dataset you have would let you quantify the value of using longer windows, etc. I think at least showing some concrete comparison of the current window vs a 7 day window would be needed.
> >
> > I looked thru the plots in J, I'm not too convinced the current solution is "good enough" and would not be substantially improved with day-of-week effects.
> >
> > RE baselines with nearest neighbors
> >
> > One lingering issue: it seems weird to use K=3 for your method (which seems to consistently do best), but only let the baseline use K=1 or K=9 or higher.
> >
> > RE outliers
> >
> > I don't think just using the sigmoid to rescale things solves my concern... even if there is some lone data point way far from the existing "normal" data, your approach will still force its predictions to be based off (potentially very unrelated) neighbors, rather than say falling back on some simpler heuristic.
> >
> > I am happy that there is some experimental evidence that the method is somewhat robust to outliers, though.
> >
> > I think I'd like to see a more controlled experiment, where there is some outlier with opposite response to any neighbor, and we see how its prediction changes as its distance grows (and how this differs from simpler approaches).

---

> > > ### Author Response · Authors · 2019-11-15
> > > **Thanks for your reply, more experiments were added**
> > >
> > > Thanks very much for taking time to read our response. We have updated the paper to address your concerns, as follows:
> > >
> > > Q: I find it a bit hard to believe that storing (sparser) traffic data long enough to capture day of week effects wouldn't be worth it.  I think at least showing some concrete comparison of the current window vs a 7 day window would be needed.
> > >
> > > A: We wish to further clarify that in our previous response, we mean that using shorter input to perform long-term forecasting is more desirable, given the overhead incurred by data measurements, as suggested by [1]. That is not to say that exploiting periodic information may not help improve the forecasting performance. However, recall that a 7-day window corresponds to a 2016-long sequence as input, since the data is sampled every 5 minutes. It is very difficult for RNN-based models to handle such long sequences. In addition, by considering the number of mobile services (38) and antennas (792), the input for 7 days would have 60,673,536 data points. This would make any forecasting model extremely large and therefore impractical for real deployment.
> > >
> > > To capture seasonal information more efficiently, we concatenate the 30 minute-long sequences (sampled every 5 minutes) with a sub-sampled 7-day window (sampled every 2h). This forms an input with length 90 (6 + 84). We conduct experiments on a subset of the mobile traffic dataset (City 1) and show the results in the Appendix M of the revised paper. The results suggest that the performance can indeed be improved by introducing seasonal information, and we believe this is a promising direction that we can further explore in future work.
> > >
> > > However, and more importantly to the present study, we also remark that the improvement determined by considering a richer input is fairly uniform across models (see Table~7 in the paper): therefore, our conclusion that CloudLSTM outperforms state-of-the-art benchmarks in the mobile traffic forecasting task still holds.
> > >
> > > [1] Chaoyun Zhang and Paul Patras. "Long-term mobile traffic forecasting using deep spatio-temporal neural networks." Proc. ACM MobiHoc. 2018.
> > >
> > > Q: It seems weird to use K=3 for your method (seems best), but only let the baseline use K=1 or K=9 or higher.
> > >
> > > A: K=3 was not shown for the baselines, because we did not see clear performance improvements in their performance with different K values. To eliminate any doubt, we added results with K = {3 ,6} for each baselines and updated Table 6 in the revised paper (note the numbering has changed).
> > >
> > > Q: Even if there is some lone data point way far from the existing "normal" data, your approach will still force its predictions to be based off (potentially very unrelated) neighbors, rather than say falling back on some simpler heuristic. I am happy that there is some experimental evidence that the method is somewhat robust to outliers, though.
> > >
> > > A: Thanks for acknowledging our efforts to how the robustness of our CloudLSTM to outliers. We want to emphasize that the DConv is only an operator of the model and it relies on neighbors of each point. Therefore, only taking one DConv layer for the forecasting might indeed force its predictions over outliers to be based on unrelated neighbors. This is because one DConv will not have sufficient representability to learn proper weights individually for inliers and outliers. However, the DConv is only a component of the entire architecture. By stacking multiple such components (DConv) via dedicated structure (LSTM), the CloudLSTM has much stronger representability and therefore it will be able to learn appropriate weights for outliers between target points and their neighbors.
> > >
> > > This is similar to convolutional operator over images. Though one Conv operator only sums the shared weights over each anchor point and its neighbors, a deep stack of CNN can learn very complex correlations within pixels, and therefore performs remarkably in many Computer Vision applications. Our experiments in Appendix K clearly show that the performance of outliers will not be affected by their "lone" positions.
> > >
> > > Q: I think I'd like to see a more controlled experiment, where there is some outlier with opposite response to any neighbor, and we see how its prediction changes as its distance grows.
> > >
> > > A: To further show that our CloudLSTM is robust to outliers, we re-run the experiments on the air quality dataset, where we push some weather stations away from the center with different distances, so as to construct artificial outliers. Our experiments show that our CloudLSTM performs equally well when forecasting over inliers and outliers, irrespective of the moving distance of the outliers. Importantly, CloudLSTM achieves significantly better performance over its counterpart PointLSTM (PointCNN + LSTM). New results can be found in Appendix K of the revised paper, and the synthetic dataset will be also released publicly for the sake of reproducibility.

---

> ### Author Response · Authors · 2019-11-10
> **Response to Blind Review #1 (2)**
>
> Continues  from the `````````````“Response to Blind Review #1 (1)”
>
> Q: It's not clear to me that the experiments here consider realistic scenarios.  Certainly there are time-of-day, day-of-week, and seasonal effects that are all important. At a minimum, I'd think that for the mobile traffic case you could at least look at consuming the last 48 hr of data and predicting the next 30-90 minutes. I suspect that would make even simpler models do much better.
>
> A: There are several reasons for not using long sequences as input: (1) Data measurements are expensive; while measuring air quality is easier, mobile data collection is not straightforward, as it relies on dedicated hardware and involves substantial data processing overhead. Therefore, mobile traffic collection is not activated all the time and thus we may not obtain full 24-48 hours of data for making predictions [1]. Therefore, relying on short-term input to make long-term performance predictions is in fact more realistic [2]; (2) The time complexity for RNN-based models grows linearly with the length of the input. In our case, the mobile traffic dataset is sampled every 5 minutes and a 48-hour data trace would correspond to 576-long sequence, which will hard for RNN-based models to handle; (3) By looking at the heat map of the forecasting in Figs 11 and 12 in Appendix J.2, we see that the CloudLSTM is already performing well, given short time series as input; (4) We agree that including periodic information may improve performance; instead of increasing the length of the input, a smarter way is to mix that with the predictions made [2] or taking those as different channels of the input, as they do not increase complexity significantly. This is an avenue we will pursue as part of future work.
>
> [1] Chaoyun Zhang, Xi Ouyang, and Paul Patras. "ZipNet-GAN: Inferring fine-grained mobile traffic patterns via a generative adversarial neural network." Proc. ACM International Conference on emerging Networking EXperiments and Technologies. 2017.
>
> [2] Chaoyun Zhang and Paul Patras. "Long-term mobile traffic forecasting using deep spatio-temporal neural networks." Proc. ACM International Symposium on Mobile Ad Hoc Networking and Computing. 2018.\\
>
>
> Q: I think the experiments are missing some key simple baselines (or I misunderstand something). For example, rather than the complicated CNN/LSTM architectures, why not try to directly see how much value there is in "neighbors" in this 2d space? At each point, you can make predictions using only the K nearest neighbors' data, with K swept from 1 to 100 or something. I would expect with these features, using just a simple MLP or RNN would do quite well. I'd like to see a stronger qualitative case made for why we expect the complicated D-Conv weighting operator here to do better than this baselines.
>
> A: We agree such baselines should be used for comparison, therefore in the revised manuscript we consider them in Appendix L, where we show the performance of MLP and LSTM (which only take a sequence of k-nearest neighbors of each point as input). We test with K in \{1, 9, 25, 50, 100\}, and find that K does not affect the forecasting performance significantly. Importantly, our CloudLSTM achieves much better performance than these baselines.
>
>
> Q: Overall, the results tables appear promising (for app traffic forecasting in Table 1, the proposed CloudLSTM achieves 3.66 MAE compared to 4.95 for PointCNN and 4.8 for an MLP). However, it's not clear why and I'd like to understand why. Is it that the other approaches are overfitting?
>
> A: Our CloudLSTM is able to learn dynamic spatial correlations in point-clouds through different time steps, while the others does not have this nice property. According to our reply to Reviewer 2, it appears that the D-Conv contributes most to the performance, followed by the RNN, and the attention mechanism. This is not because other approaches are overfitting. We add such a discussion in Sec. 4.4 of the revised paper.
>
>
> Q: I would suggest avoiding calling the method "$\mathcal{D}$-Conv", and instead use just "DConv", since this is easier to type into search engines and easier to search for in a PDF document
>
> Related: Point clouds could be represented as graphs, and then use graph embeddings as feature representations
>
> A: Thanks for the suggestion. We change $\mathcal{D}$-Conv to DConv in the revised paper. Regarding the graph embedding, we agree this will have potential and will consider this for future work.

---

### Official Review · AnonReviewer2 · 2019-10-28
**Official Blind Review #2**

**Rating:** 3

**Review:**

The paper proposed a new convolution operator, named dynamic post-cloud convention over spatiotemporal data, and the convolution operator can be embedded in different neural network architectures, like recurrent neural networks. In order to achieve the convolution over point-clouds by using both value features and the spatial-features, given a data point, the convolution is conducted over its k-nearest neighbors generated by CNN.  They compared the proposed convolution method by embedding it into RNN, GRN, and LSTM against a number of existing methods on two datasets, in terms of MAE, RMSE, PSNR, and SSIM. Overall, this paper is interesting but needs some clarifications on

1. Given that the proposed convolution operator use KNN to choose the nearest neighbors. It would be good to empirically to study how K would affect the performance, does it data-dependent
2. Is it possible to study the time complexity for various models?
3. Table 1 and table 2 seem to show that the proposed convolution operator contributes to the performance in terms of the mean of each metric. It might be good to do further oblation test to study which mechanism actually contribute to the performance. The choice of RNN, attention, or the new operator? Furthermore, the std is quite large, which makes one wonder if the improvement is statistically significant.

**Experience Assessment:**

I have read many papers in this area.

**Review Assessment: Checking Correctness Of Derivations And Theory:**

I did not assess the derivations or theory.

**Review Assessment: Checking Correctness Of Experiments:**

I assessed the sensibility of the experiments.

**Review Assessment: Thoroughness In Paper Reading:**

I made a quick assessment of this paper.

---

> ### Author Response · Authors · 2019-11-10
> **Response to Blind Review #2**
>
> Thank you for your feedback and for indicating which aspects can be improved on. We note that some of the comments made are already addressed in the original manuscript.
>
> Q: Given that the proposed convolution operator use KNN to choose the nearest neighbors. It would be good to empirically to study how K would affect the performance, does it data-dependent?
>
> A: The impact of K (ranging from 3 to 9) is analyzed in Tables 1 and 2, which suggest that K only affects marginally the prediction performance. Regarding other baseline models based on k-nearest-neighbours (i.e., MLP and LSTM), we test with K ranging from 1 to 100 and the results obtained give a similar conclusion. These results can be found in Table 6 of Appendix L in the revised manuscript.
>
> Q: 2. Is it possible to study the time complexity for various models?
>
> A: We gave a detailed complexity analysis of our D-Conv in Appendix B, which suggests that compared to the convolution operator whose inputs, outputs, and filters have the same size, the D-Conv only introduces additional complexity by searching the $\mathcal{K}$ nearest neighbors for each point $O(\mathcal{K}\cdot L\log N)$. Such complexity does not increase much even with higher dimensional point-clouds.
>
> Q: It might be good to do further oblation test to study which mechanism actually contribute to the performance. The choice of RNN, attention, or the new operator?
>
> A: We conducted our experiments using strict variable-controlling methodology, i.e., only changing one factor while keep the remaining the same. Therefore, it is easy to study the effects of each factor. For example, taking a look at the performance of LSTM, ConvLSTM, PredRNN++, PointLSTM and CloudLSTM, which employ dense layers, and CNN, PointCNN and D-Conv as core operators but using LSTM as the RNN structure, it is clear that the D-Conv contributes significantly to the performance improvements. Further, by comparing CloudRNN, CloudGRU and CloudLSTM, it appears that CloudRNN $\ll$ CloudGRU $<$ CloudLSTM. Similarly, by comparing the CloudLSTM and Attention CloudLSTM, we see that the effects of the attention mechanism are not very significant. Therefore, we believe the core operator $>$ RNN structure $>$ attention, ranked by their contribution.
>
> Q: Furthermore, the std is quite large, which makes one wonder if the improvement is statistically significant.
>
> A: We note that the large std exists only in the Cluster A of the air quality dataset. We checked this dataset more carefully, and found that the level of noise therein is more severe than in other case studies. We believe that is the root cause of the larger std. In addition, we note that all models achieve similar std in this dataset (except for MLP), while our proposal obtains the best mean for all metrics, which proves that the improvement is statistically significant.

---

### Official Review · AnonReviewer3 · 2019-11-02
**Official Blind Review #3**

**Rating:** 8

**Review:**

=========== Update after rebuttal

Thanks for the clarifications and the update. I recommend acceptance of the paper and updated to 8.

Last comment: please still improve the appearance of Figure 4 by using a more diverse set of marker shapes as well as overlay and offset tricks -- see https://www.cs.ubc.ca/~schmidtm/Software/prettyPlot.html for an example.

============

This paper introduces a new convolution operator (D-conv) specifically tailored to model point-cloud data evolving over time, i.e. a set of n points with features and localization-coordinates that can evolve over time. The main idea is to use the k-nearest neighbor structure for each point to get a fixed size k window to use in the convolution to determine the new location and feature values of a point (and a permutation-invariant operation). The D-Conv operator is included in a LSTM architecture (CloudLSTM) to enable the spatio-temporal modeling of point-cloud data, and can be combined in standard neural network architectures such as a Seq2Seq with attention. This is in contrast to previous approaches which modeled the data on a grid through preprocessing, or did not include the temporal component for cloud data. Experiments is conducted on 4 benchmark datasets, covering two point-cloud stream forecasting applications, showing how CloudLSTM give lower prediction error than numerous baselines and alternatives.

While the D-Conv idea seems fairly simple and natural, it is novel AFAIK and fairly appropriate to model point-cloud data streams. The approach is well situated in the literature, and the experiments are indicative that this method can improve on the current approaches. I am thus leaning towards accept.

The paper is fairly clear, though the notation is a bit confusing and somewhat sloppy (see detailed comment below).

Important clarification requested: the current notation suggests that each channel could have a different location for a point p_n, the K nearest points seem to be defined irrespective on the channel. So is the location fixed across channels; or does this paper allow the neighborhood structures to vary across channel?

== Other detailed comments ==

- p.3 Q_n^K -- it seems it would be more appropriate to define it as an ordered list of k points (rather than a set, as this would loose all information about the order); unless you append a new dimension to each point where you put the ordering information there for the purpose of defining the k points in Q_n^K.

- (2) the notation is a bit weird and overloaded for the summation (without being defined). Examples include "i in U1" (when U1 is an integer, not a set); "p_n^k in Q_n^K" when p_n^k does not appear in the summation (a clearer alternative would be using the notation v(p_n^k)_i^h for the h^th value of channel i of point p_n^k, e.g.; now p_n^k would indeed appear in the expressoin); "v_n^h in v_n" -> why not just summing over h as it is really doing? Etc.!

- (2) S_out^j: each p_n^' should be a *tuple* (not a set like currently written).

- Figure 4: the lines are really hard to distinguish just by the similar colors -- please use different markers for the different lines (and offset the marker so that they can be seen)

- Several neighborhood sizes are experimented with. Note though that smaller neighborhood sizes are just *special cases* of bigger neighborhood sizes (by using zero weight on the last few neighbors in the convolution). Wouldn't it make sense to use a big neighborhood size and regularize in some way the weights for the further neighbors?

- Table 2: for SSIM, there are two rows with 0.69 +/- 0.07 (minimal value) -- they could be both bolded.

- Appendix B, they claim that the complexity of finding the K nearest neighbors (in dimension L for n points) is close to O(K L log(n)) if using KD trees. I vaguely recall issues in high dimension though (in particular that the above complexity is only valid for specific distributions of points in low dimension). E.g. see https://en.wikipedia.org/wiki/K-d_tree#High-dimensional_data where it is mentioned that L << log(n) is normally needed to guarantee efficiency. The claim should properly be nuanced.

**Experience Assessment:**

I have published one or two papers in this area.

**Review Assessment: Checking Correctness Of Derivations And Theory:**

I assessed the sensibility of the derivations and theory.

**Review Assessment: Checking Correctness Of Experiments:**

I assessed the sensibility of the experiments.

**Review Assessment: Thoroughness In Paper Reading:**

I read the paper at least twice and used my best judgement in assessing the paper.

---

> ### Author Response · Authors · 2019-11-10
> **Response to Blind Review #3**
>
> We appreciate the detailed and positive comments, which clearly reflect many of the essential contributions of our work. We answer the points raised in what follows.
>
> Q: Important clarification requested: the current notation suggests that each channel could have a different location for a point $p_n$, the K nearest points seem to be defined irrespective on the channel. So is the location fixed across channels; or does this paper allow the neighborhood structures to vary across channel?
>
> A: The K nearest points can indeed vary for each channel at each location. The reason is that the channels in the point-cloud dataset may represent different types of measurements. For example, channels in the mobile traffic dataset are related to the traffic consumption of different mobile apps, while those in the air quality dataset are different air quality indicators (SO2, CO, etc.). The spatial correlations will vary between different measurements (channels), due to human mobility. For instance, more people may use Facebook at a social event, but YouTube traffic may be less significant in this case. This will be reflected by the data consumption of each app. The same applies to air quality indicators affected by vehicle movement and factory working times. We want these spatial correlations to be learnable, so we do not fix the K nearest neighbors across channels, but encourage each channel to find the best neighbor set. This is also a contribution of the CloudLSTM, which helps improve the forecasting performance. We add this discussion in Sec. 3.2 of the revised paper.
>
> Q:- p.3 $Q_n^K$ -- it seems it would be more appropriate to define it as an ordered list of k points (rather than a set, as this would loose all information about the order); unless you append a new dimension for the defining the k points in $Q_n^K$.
>
> - (2) the notation is a bit weird and overloaded for the summation (without being defined). Examples include "i in U1" (when U1 is an integer, not a set); "$p_n^k$ in $Q_n^K$" when $p_n^k$ does not appear in the summation (a clearer alternative would be using the notation $v(p_n^k)_i^h$ for the $h^{th}$ value of channel i of point $p_n^k$, e.g.; now $p_n^k$ would indeed appear in the expression); "$v_n^h$ in $v_n$"  why not just summing over h as it is really doing? Etc.!
>
> - (2) $S_{out}^j$: each $p_n'$ should be a tuple (not a set like currently written).
>
> A: We very much appreciate the detailed comments on these issues. We have update notation accordingly in the revised manuscript, which is available online via OpenReview. We trust the revision eliminates any confusion.
>
> Q:  Figure 4: the lines are really hard to distinguish just by the similar colors -- please use different markers for the different lines.
>
> A: Thank you for raising this issue. We acknowledge the original figure had readability issues and have updated it (and Fig. 9) to address this problem.
>
> Q:  Several neighborhood sizes are experimented with. Note though that smaller neighborhood sizes are just special cases of bigger neighborhood sizes. Wouldn't it make sense to use a big neighborhood size and regularize in some way the weights for the further neighbors?
>
> A: This is an excellent point. We agree that smaller neighborhood sizes are special cases of bigger neighborhood sizes. The reasons we do not use a large K are because (1) as analyzed in Appendix B, the complexity of D-Conv grow linearly with K; (2) While testing with K ranging between 3 and 9, we do not see clear performance improvements with higher K; (3) The K is equivalent to the receptive field of a normal CNN kernel (e.g., K=9 is is equivalent to a 3*3 CNN kernel), and a small CNN kernel has been proven effective in imaging applications. To answer Reviewer 1's question, we test with K between 1 and 100 for k-nearest-neighbor based MLP and LSTM in Appendix L; the results also suggest a large K does not improve performance. At the same time, we agree that using a big neighborhood size and regularizing in some way may be appropriate, as it may reduce overfitting caused by large K. We will consider this approach for future work.
>
> Q: Table 2: for SSIM, there are two rows with 0.69 +/- 0.07 (minimal value) -- they could be both bolded.
>
> A: This has been revised. Thanks.
>
> Q: Appendix B, they claim that the complexity of finding the K nearest neighbors (in dimension L for n points) is close to O(K L log(n)) if using KD trees. I vaguely recall issues in high dimension though (in particular that the above complexity is only valid for specific distributions of points in low dimension).  where it is mentioned that L $<<$ log(n) is normally needed to guarantee efficiency. The claim should properly be nuanced.
>
> A: This is correct, and, in fact, in real life the dimensions of a point-cloud dataset are normally 2 or 3. Also, we usually have much more than 3 points in the dataset. Therefore, L $<<$ log(n) should hold for most applications. We clarified this aspect as a footnote in Appendix B.

---

### Author Response · Authors · 2019-11-11
**Paper has been revised**

We appreciate the valuable feedback from all reviewers.  We have revised the manuscript to address the concerns

---

### Decision · Program_Chairs · 2019-12-19

**Decision:**

Reject

**Comment:**

The paper presents an approach to forecasting over temporal streams of permutation-invariant data such as point clouds. The approach is based on an operator (DConv) that is related to continuous convolution operators such as X-Conv and others. The reviews are split. After the authors' responses, concerns remain and two ratings remain "3". The AC agrees with the concerns and recommends against accepting the paper.